# AN UNLEARNING-ENHANCED GENERAL FRAMEWORK FOR TEST-TIME ADAPTATION

## ABSTRACT

Test-time Adaptation (TTA) aims to mitigate performance degradation caused by distribution shifts during testing time. While various TTA approaches exist, such as entropy minimization, pseudo-labeling, weight-space regularization and Bayesian methods, a generalized optimization framework for TTA is currently absent. To address this gap, we present a general framework for TTA. This framework provides a conceptual basis for understanding existing methods as specific instances within a broader optimization framework, and facilitates the development of new TTA methods. Additionally, our proposed framework brings attention to limitations in existing approaches by unveiling an implicit assumption that all source domain knowledge is universally beneficial for adapting to the target domain. In reality, only a portion of the source domain knowledge is useful due to potential large distribution discrepancies between the source and target domains. Based on this insight, we build upon our general framework and derive a novel method named UnLearning-enhanced test-time adaptation (*Lana*). Specifically, it adaptively unlearns irrelevant source domain knowledge and then adapts to the target test domain. Through thorough theoretical analysis and empirical results, we showcase the effectiveness of our proposed method in enhancing TTA performance. This work contributes not only a broader understanding of TTA through a general framework but also a novel practical solution, *Lana*, derived from our general framework, offering a foundation for further advancements in addressing distribution shifts during testing in machine learning models.

## 1 INTRODUCTION

Deep neural networks demonstrate optimal performance when both the training and testing data conform to identical distributions. Nonetheless, this assumption does not align with the reality of real-world applications, where the distribution of test data frequently diverges from that of the training data. This incongruity inevitably results in a noticeable decline in performance when deploying a pre-trained model on such divergent test distributions. Consequently, the imperative arises to adapt the pre-trained model to the test data distribution (domain) in real-world applications, thereby mitigating the disparity between the training and test data.

To tackle this challenge, the concept of test-time adaptation (TTA) (Wang et al., 2021) is introduced, which involves adjusting a pre-trained model through the creation of a loss function solely based on unlabeled test data. Various TTA approaches have emerged, encompassing: (1) Entropy-minimization-based methods, such as Tent (Wang et al., 2021) and SAR (Niu et al., 2023); (2) Pseudo-labeling-based methods, including TAST (Jang et al., 2023) and AdaContrast (Chen et al., 2022a); (3) Weight-regularization-based methods, like EATA (Niu et al., 2022) and SWR (Choi et al., 2022). (4) Output-regularization-based methods: LAME (Boudiaf et al., 2022); and (5) Bayesian methods: SSA (Lee, 2025). Despite the diverse range of available methods, designing TTA approaches often necessitates substantial intuition. For example, EATA (Niu et al., 2022) and SWR (Choi et al., 2022) need carefully designed regularizations. Moreover, a general framework is currently absent to place existing methods within a shared optimization objective, as well as to offer guidance for the development of novel TTA methods.

In an effort to bridge this gap, we propose a general and versatile optimization objective for TTA. This framework offers several advantageous outcomes. First, many existing TTA methods can be

Table 1: A general optimization framework for TTA. We define a generalized TTA optimization objective as $\mathcal{L}^{TTA} = \alpha D_{\boldsymbol{\Phi}}(g_{\boldsymbol{\theta}}(\boldsymbol{x}), \boldsymbol{z}) + \beta D_{\boldsymbol{\Psi}}(\boldsymbol{\theta}, \boldsymbol{\theta}_*) \pm \mathcal{L}_{CE}(\mathcal{D}_{id}, \boldsymbol{\theta}_*)$. Where $\alpha, \beta$ are constants, $g_{\boldsymbol{\theta}}(\boldsymbol{x})$ denotes the output class probabilities on a test data $\boldsymbol{x}$, $D_{\boldsymbol{\Phi}}(g_{\boldsymbol{\theta}}(\boldsymbol{x}), \boldsymbol{z})$ is *output space* regularization represented as a Bregman divergence associated with function $\boldsymbol{\Phi}$, $D_{\boldsymbol{\Psi}}(\boldsymbol{\theta}, \boldsymbol{\theta}_*)$ is *weight space* regularization represented as a Bregman divergence associated with function $\boldsymbol{\Psi}$, $\mathcal{L}_{CE}(\mathcal{D}_{id}, \boldsymbol{\theta}_*)$ is the cross-entropy loss on in-distribution (ID) data. This last loss term is optional for TTA, but may be present in some TTA methods. Various TTA methods can be recovered from this general optimization objective by setting different $\boldsymbol{\Phi}$, $\boldsymbol{\Psi}$ and the arguments of the Bregman divergence.

| Category | Method | Recover Setting |
|---|---|---|
| Entropy-minimization | Tent (Wang et al., 2021) | $\alpha = -1, \beta = 0, \boldsymbol{\Phi}(\boldsymbol{p}) = \sum_{i=1}^{i=n} \boldsymbol{p}_i \log \boldsymbol{p}_i \quad \boldsymbol{p} = g_{\boldsymbol{\theta}}(\boldsymbol{x})$ |
| | SAR (Niu et al., 2023) | $\alpha = -1, \beta = 0, \boldsymbol{\Phi}(\boldsymbol{p}) = \sum_{i=1}^{i=n} \boldsymbol{p}_i \log \boldsymbol{p}_i \quad \boldsymbol{p} = g_{\boldsymbol{\theta}+\varepsilon(\boldsymbol{\theta})}(\boldsymbol{x})$ |
| | | $\varepsilon(\boldsymbol{\theta}) = \rho \, \text{sign}(\nabla_{\boldsymbol{\theta}} \mathcal{L}_{\boldsymbol{\theta}}^{TTA}) |\nabla_{\boldsymbol{\theta}} \mathcal{L}_{\boldsymbol{\theta}}^{TTA}| / ||\nabla_{\boldsymbol{\theta}} \mathcal{L}_{\boldsymbol{\theta}}^{TTA}||_2$ |
| Pseudo-labeling | TAST (Jang et al., 2023) | $\alpha = 1, \beta = 0. \; \boldsymbol{\Phi}(\boldsymbol{p}) = \sum_{i=1}^{i=n} \boldsymbol{p}_i \log \boldsymbol{p}_i. \; D_{\boldsymbol{\Phi}}(\boldsymbol{p}, \boldsymbol{q}) = \mathbb{KL}(g_{\boldsymbol{\theta}}(\boldsymbol{x}), \hat{g}_{\boldsymbol{\theta}}(\boldsymbol{x}))$ |
| | AdaContrast (Chen et al., 2022a) | $\alpha = 1, \beta = 0. \; \boldsymbol{\Phi}(\boldsymbol{p}) = \sum_{i=1}^{i=n} \boldsymbol{p}_i \log \boldsymbol{p}_i. \; D_{\boldsymbol{\Phi}}(\boldsymbol{p}, \boldsymbol{q}) = \mathbb{KL}(g_{\boldsymbol{\theta}}(\boldsymbol{x}), \boldsymbol{y})$ |
| Weight-regularization | EATA (Niu et al., 2022) | $\alpha = -1, \boldsymbol{\Psi}(\boldsymbol{\theta}) = \frac{1}{2} \boldsymbol{\theta}^T F \boldsymbol{\theta}, \; F$ is a diagonal Fisher information matrix |
| | SWR (Choi et al., 2022) | $\alpha = -1, \boldsymbol{\Psi}(\boldsymbol{\theta}) = \frac{1}{2} \boldsymbol{\theta}^T M \boldsymbol{\theta}, \; M$ is a diagonal sensitivity matrix |
| Output-regularization | LAME (Boudiaf et al., 2022) | $D_{\phi}(\tilde{\boldsymbol{z}}_i, \boldsymbol{q}_i) = \text{KL}(\tilde{\boldsymbol{z}}_i \parallel \boldsymbol{q}_i)$ and $D_{\phi}(\boldsymbol{z}, \boldsymbol{z}') = \frac{1}{2}\|\boldsymbol{z} - \boldsymbol{z}'\|^2$ |
| Bayesian-based | SSA (Lee, 2025) | $D_{\boldsymbol{\Phi}}(\boldsymbol{p}, \boldsymbol{q}) = \mathbb{KL}(g_{\boldsymbol{\theta}}(\boldsymbol{x}), \boldsymbol{v}).$ |
| | | $\boldsymbol{p} = \boldsymbol{\theta}, \boldsymbol{q} = \boldsymbol{\theta}_*, D_{\boldsymbol{\Psi}}(\boldsymbol{p}, \boldsymbol{q}) = (\boldsymbol{\theta} - \boldsymbol{\theta}_*)^T \boldsymbol{\Lambda}_k (\boldsymbol{\theta} - \boldsymbol{\theta}_*)$ |
| Our method | *Lana* | $\boldsymbol{\Phi}(\boldsymbol{p}) = \sum_{i=1}^n \boldsymbol{p}_i \log \boldsymbol{p}_i.$ First-order Taylor expansion to the first term. |
| | | Second-order Taylor expansion to the third term. |
| | | (We provide the derivation details in the Framework and Method Section.) |

easily *reinvented* by our general framework with minimal effort, offering deeper insights into their shared characteristics. Secondly, researchers and practitioners can circumvent redundant efforts and expedite the development of new TTA methods by leveraging our general framework as a foundation. As detailed in Table 1, our framework is designed to flexibly accommodate and recover a diverse collection of TTA methods across different categories by setting different Bregman divergence (Banerjee et al., 2005). In addition, our framework reveals that existing TTA methods only emphasize direct adaptation of the pre-trained model to the target test data distribution. However, test data may significantly differ from training data. Accordingly, the knowledge learned in pre-trained models may interfere with the test data distribution. As a result, directly adapting the pre-trained model to the target test data distribution could result in negative transfer effects (Zhang et al., 2022b), potentially leading to suboptimal performance for the target test data distribution.

To address this challenge, we build upon our general framework and introduce an innovative approach named un**L**earning-enh**an**ced test-time **a**daptation (*Lana*) to boost the effectiveness of TTA. Specifically, *Lana* is derived from our general TTA framework and it consists of a two-step process: initially, we employ adaptive unlearning to remove irrelevant and less important information from the pre-trained source domain weights. Subsequently, we adapt the unlearned model to the distribution of the target test data. This approach draws inspiration from two sources. First, it is biologically-inspired and takes cues from human learning, where the process of unlearning plays a significant role in acquiring new skills (Gravitz, 2019; Wang et al., 2025), aligning with insights from neuroscience that highlight the importance of unlearning in cognitive processes and learning new knowledge (Davis & Zhong, 2017; Richards & Frankland, 2017). Second, it addresses the common issue in neural networks where the pre-trained neural network tends to easily memorize irrelevant and unimportant information in source domain training data (Carlini et al., 2019), which hampers their adaptability to new, unseen target domain data, as their model capacity becomes cluttered with irrelevant and unimportant source domain information (Feldman & Zhang, 2020). Our proposed *Lana* is a new TTA paradigm and can be integrated with existing TTA methods to further improve their performance. To highlight the disparities between conventional TTA methods and our approach, *Lana*, we depict their distinctions with *Lana* in Figure 1.

To evaluate the effectiveness of the proposed method, we perform extensive experiments on two large-scale and challenging TTA datasets, (1) ImageNet-C which consists of various TTA scenarios with imbalanced data labels, mixture of different test data distributions and small batch size; (2) DomainNet which consists of various natural data distribution shift. In particular, when integrating the proposed adaptive unlearning with the Tent method (Wang et al., 2021), our method substantially improves the TTA performance from 47.3% to 61.1% with VitBase backbone and improves the performance from 22.0% to 37.4% with ResNet50 backbone on ImageNet-C, indicating the significant benefits of

adaptive unlearning. Furthermore, the results show that our method substantially outperforms those state-of-the-art (SOTA) TTA methods by more than 3%.

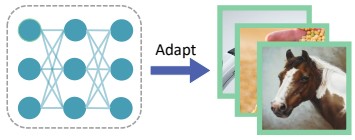 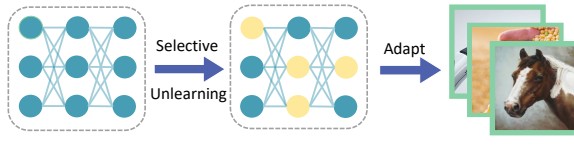

(a) Existing TTA methods               (b) Lana (Ours)

Figure 1: Comparisons between Existing TTA methods and Lana (Ours). (a) Conventional TTA methods operate under the assumption that all source-domain knowledge is universally beneficial for the target domain, leading them to directly adapt the pre-trained models from the source domain to the target domain. (b) However, in practical scenarios, the substantial dissimilarity between the source and target domains renders only a fraction of the source-domain knowledge pertinent for effective adaptation to the new domain. *Lana* (Our approach) employs a more nuanced strategy: it adaptively discards less relevant source-domain knowledge before adapting the model to the target domain. By strategically unlearning unimportant information, *Lana* optimizes the adaptation process for enhanced performance on test domains.

Our contributions can be summarized as the following:

- We introduce a general optimization framework for TTA, incorporating entropy-minimization, pseudo-labeling, weight-regularization, output-regularization and Bayesian methods. Additionally, the framework offers a general guideline for the development of novel TTA methods.

- Building upon our proposed TTA framework, we develop and derive a novel biologically-inspired unlearning-enhanced TTA method aimed at enhancing adaptability to target test data distributions.

- Extensive theoretical analysis and experiments conducted on large-scale TTA datasets validate the effectiveness of the proposed method.

## 2 RELATED WORK

### 2.1 TEST-TIME ADAPTATION

Test-time adaptation (TTA) (Liang et al., 2020; Schneider et al., 2020; Wang et al., 2021; Iwasawa & Matsuo, 2021; Mummadi et al., 2021; Zhou & Levine, 2021; Sun et al., 2020; Liu et al., 2021; Bartler et al., 2022; Gandelsman et al., 2022; Wang et al., 2022; Gong et al., 2022; Boudiaf et al., 2022; Gao et al., 2022; Kim et al., 2022; Shu et al., 2022; Goyal et al., 2022; Zhang et al., 2022a; Shin et al., 2022; Yuan et al., 2023; Lim et al., 2023; Zhao et al., 2023; Zhou et al., 2023; Kang et al., 2023; Prabhudesai et al., 2023; Peng et al., 2023; Brahma & Rai, 2023; Yu et al., 2023; Lee, 2025) refers to the process of adjusting or fine-tuning a pre-trained model on unseen target test data distribution during the testing phase. The goal is to enhance the model's performance on test data, especially when the test data distribution differs from the training data distribution.

**Fully Test-Time Adaptation** Fully TTA can adapt to the target test data distribution without changing the training procedure. Fully TTA methods can be further categorized into: (1) *entropy minimization-based methods* which minimize the model prediction entropy on the target test data distribution and then make predictions on test data distribution, consisting of Tent (Wang et al., 2021) and SAR (Niu et al., 2023); (2) *pseudo-labeling-based methods* leverage a pre-trained model to predict the target test data, generating pseudo-labels that are subsequently employed to compute the adaptation loss, including TAST (Jang et al., 2023) and AdaContrast (Chen et al., 2022a); (3) *weight-regularization-based methods* achieve TTA by applying regularization to the pre-trained model weights, ensuring that different model weights undergo updates with varying learning rates, including EATA (Niu et al., 2022) and SWR (Choi et al., 2022). However, there is currently a lack of a general optimization framework to understand these different approaches and a general guideline for creating potentially

novel TTA methods. On the other hand, existing TTA methods primarily concentrate on the direct adaptation of source domain knowledge to the target test data distribution. But, they often neglect the fact that not all source domain knowledge is relevant or beneficial for the target test data distribution, which can significantly differ from the training data distribution. In contrast, our approach prioritizes the adaptive unlearning of unimportant or irrelevant source domain knowledge from the pre-trained model, enabling a more effective adaptation to the characteristics of the target test data distribution.

## 2.2 Machine Unlearning

Machine unlearning (MU), as discussed in works such as (Guo et al., 2020; Wu et al., 2020; Bourtoule et al., 2021; Ullah et al., 2021), involves the deliberate removal or erasure of previously acquired information or knowledge from a pre-trained model. This practice is particularly relevant in the context of adhering to privacy regulations (Ginart et al., 2019). The existing approaches in MU can be further categorized into: (1) *Exact Unlearning*: This approach achieves the same effect as retraining from scratch with the remaining dataset. Representative works include (Wu et al., 2020; Bourtoule et al., 2021; Sekhari et al., 2021; Ullah et al., 2021). However, exact unlearning is computationally and memory inefficient to achieve. (2) *Approximate Unlearning*: This approach aims to improve unlearning efficiency by reducing the requirement of exact unlearning. Representative works include (Guo et al., 2020; Nguyen et al., 2020; Mehta et al., 2022).

Unlike traditional MU that aim to completely erase data traces from pre-trained models, our unlearning-enhanced TTA method is designed to dynamically eliminate less relevant information from the pre-trained model. This adaptive unlearning approach significantly improves the model's adaptation ability and performance on new test domains.

## 3 Framework and Method

In this section, we outline the TTA problem setup. Subsequently, we introduce a general framework for TTA. Following that, we derive a novel TTA method from our framework.

### 3.1 Problem Setup and Preliminary

**Test-Time Adaptation** We commonly assumed that the test data $\mathcal{D}_{test}$ will exhibit the same distribution as training data. However, it is frequently observed that the distribution of test data differs from that of the training data. To tackle this challenge, TTA entails adjusting a pre-trained model using unlabeled testing data $x \sim \mathcal{D}_{test}$ using an unsupervised adaptation loss function. Afterward, the adapted model employs the updated parameters to make predictions on the test input $x$.

**Bregman Divergence** Suppose $\mathbf{\Phi} : \Omega \to \mathbb{R}$ is a continuously differentiable and strictly convex function which is defined on a convex set $\Omega$. The Bregman divergence (Banerjee et al., 2005) associated with $\mathbf{\Phi}$ for two points $p$ and $q$ can be interpreted as the difference between the $\mathbf{\Phi}$ value at point $p$ and the value obtained by approximating $\mathbf{\Phi}$ through a first-order Taylor expansion centered at point $q$, followed by the evaluation of this approximation at point $p$ as:

$$D_{\mathbf{\Phi}}(p, q) = \mathbf{\Phi}(p) - \mathbf{\Phi}(q) - \langle \nabla\mathbf{\Phi}(q), p - q \rangle \tag{1}$$

$\nabla\mathbf{\Phi}(q)$ represents the gradient of $\mathbf{\Phi}$ at point $q$, and $\langle,\rangle$ denotes the dot product between two vectors. In the following, we will utilize Bregman divergence to establish a general framework for TTA.

### 3.2 A General Optimization Framework for TTA

In the following, we reformulate and recast various established TTA algorithms in terms of a more general TTA optimization objective as the following:

$$\mathcal{L}^{TTA} = \alpha \underbrace{D_{\mathbf{\Phi}}(g_{\boldsymbol{\theta}}(\boldsymbol{x}), \boldsymbol{z})}_{\text{output space}} + \beta \underbrace{D_{\mathbf{\Psi}}(\boldsymbol{\theta}, \boldsymbol{\theta}_*)}_{\text{weight space}} \pm \underbrace{\mathcal{L}_{CE}(\mathcal{D}_{id}, \boldsymbol{\theta}_*)}_{\text{ID data (optional)}} \tag{2}$$

where $\boldsymbol{\theta}$ are the current model parameters. $\alpha, \beta$ are regularization constants. $g_{\boldsymbol{\theta}}(\boldsymbol{x})$ denotes the output class probabilities on a test data $\boldsymbol{x}$. The term $D_{\mathbf{\Phi}}(g_{\boldsymbol{\theta}}(\boldsymbol{x}), \boldsymbol{z})$ represents a form of regularization in the

*output space*. It is expressed as the Bregman divergence associated with the function $\boldsymbol{\Phi}$. The constant vector $\boldsymbol{z}$ serves as a reference vector. On the other hand, $D_{\boldsymbol{\Psi}}(\boldsymbol{\theta}, \boldsymbol{\theta}_*)$ represents a form of regularization applied to the *weight space or model parameter space*. It is also expressed as a Bregman divergence, this time associated with the function $\boldsymbol{\Psi}$. The term $\boldsymbol{\theta}_*$ refers to the optimal model parameters that were learned for source domain data. This part of loss function is used to ensure that the model doesn't adapt too rapidly (more stable) to new test domains. $\mathcal{L}_{CE}(\mathcal{D}_{id}, \boldsymbol{\theta}_*) = \mathbb{E}_{(\boldsymbol{x},y) \sim \mathcal{D}_{id}} \mathcal{L}_{CE}(\boldsymbol{x}, y)$ is the cross-entropy loss on the in-distribution (ID) data. This loss term is optional but may be required by some existing methods. For instance, the existing TTA method, EATA (Niu et al., 2022), utilizes optimization on unlabeled ID samples. EATA argues that while TTA methods do not have access to the training data, they can leverage the unlabeled ID test data. Additionally, it's worth noting that various existing TTA methods can be easily *reinvented* with this general framework. Specifically, we cast (1) entropy-minimization methods: Tent (Wang et al., 2021) and SAR (Niu et al., 2023); (2) pseudo-labeling methods: TAST (Jang et al., 2023) and AdaContrast (Chen et al., 2022a); (3) weight-regularization methods: EATA (Niu et al., 2022) and SWR (Choi et al., 2022); (4) output-regularization methods: LAME (Boudiaf et al., 2022); (5) Bayesian methods: SSA (Lee, 2025) as special instances of Eq. (2). Due to space constraints, we only outline the essential steps for deriving different TTA methods. Other details, e.g., sample selection and data augmentation, are not included since they are orthogonal to the TTA optimization, which can be integrated with them seamlessly. Due to space limitations, we present the derivations for LAME (Boudiaf et al., 2022), SWR (Choi et al., 2022), SAR (Niu et al., 2023) and SSA (Lee, 2025). We put other TTA methods, including Tent (Wang et al., 2021), EATA (Niu et al., 2022), AdaContrast (Chen et al., 2022a) and TAST (Jang et al., 2023) in Appendix. Detailed derivations can be found in Appendix B.

**SAR As A Special Case** SAR (Niu et al., 2023) is a sharpness-aware optimization (Foret et al., 2021)-based TTA method. In Eq. (2), we set $\alpha = -1, \beta = 0$ and take $\boldsymbol{\Phi}$ to be the negative entropy function, i.e., $\boldsymbol{\Phi}(\boldsymbol{p}) = \sum_{i=1}^{i=n} \boldsymbol{p}_i \log \boldsymbol{p}_i$. We set $\boldsymbol{p} = g_{\boldsymbol{\theta} + \varepsilon(\boldsymbol{\theta})}(\boldsymbol{x})$, i.e., the softmax probability output of the neural network on the test data and $\boldsymbol{q} = \boldsymbol{v}$, i.e., the uniform distribution on the class probability distribution. $D_{\boldsymbol{\Phi}}(\boldsymbol{p}, \boldsymbol{q}) = \mathbb{KL}(g_{\boldsymbol{\theta} + \varepsilon(\boldsymbol{\theta})}(\boldsymbol{x}), \boldsymbol{v})$. We then recovered the SAR method.

**SWR As A Special Case** SWR (Choi et al., 2022) is a weight-regularization-based method. It can be expressed as the following objective:

$$\mathcal{L}^{TTA} = -\mathbb{KL}(g_{\boldsymbol{\theta}}(\boldsymbol{x}), \boldsymbol{v}) + \mathbb{KL}(\hat{g}_{\boldsymbol{\theta}}(\boldsymbol{x}), \boldsymbol{v}) + \beta(\boldsymbol{\theta} - \boldsymbol{\theta}_*)^T M(\boldsymbol{\theta} - \boldsymbol{\theta}_*) \tag{3}$$

where $M$ is a diagonal matrix, where $M = diag(m_1, m_1, \cdots, m_2, m_2, \cdots, m_L, m_L, \cdots)$. The diagonal elements in $M$ are layer-wise penalty constants which indicates how fast those layer parameters should be updated. In Eq. (2), we set $\alpha = -1$ and take $\boldsymbol{\Psi}(\boldsymbol{\theta}) = \frac{1}{2}\boldsymbol{\theta}^T M \boldsymbol{\theta}$. We set $\boldsymbol{p} = \boldsymbol{\theta}$ and $\boldsymbol{q} = \boldsymbol{\theta}_*$. $D_{\boldsymbol{\Psi}}(\boldsymbol{p}, \boldsymbol{q}) = (\boldsymbol{\theta} - \boldsymbol{\theta}_*)^T M(\boldsymbol{\theta} - \boldsymbol{\theta}_*)$. Then, we recovered the SWR.

**LAME As a Special Case** LAME (Boudiaf et al., 2022) is an output-regularization-based approach. In Eq. (2), we set $\alpha = 1, \beta = -1$. $D_{\phi}(\tilde{\boldsymbol{z}}_i, \boldsymbol{q}_i) = \mathbb{KL}(\tilde{\boldsymbol{z}}_i \parallel \boldsymbol{q}_i)$ and $D_{\phi}(\boldsymbol{z}, \boldsymbol{z}') = \frac{1}{2}\|\boldsymbol{z} - \boldsymbol{z}'\|^2$

**SSA As a Special Case** SSA (Lee, 2025) is a Bayesian-based approach.

In Eq. (2), we set $\alpha_k = \sqrt{\frac{\sigma_\lambda^2}{\eta^2 \sigma_k^2}}$, where $\sigma_\lambda$ denotes a constant associated with steady-state regime, $\eta$ is a constant, $\sigma_k$ is the standard deviation at step $k$. The posterior mean can be approximately solved by the following optimization:

$$J(\boldsymbol{\theta}) = \underbrace{\alpha_k \, \mathbb{E}_{x \sim D_k}\big[\mathbb{KL}\big(p_{\boldsymbol{\theta}}(\cdot \mid \boldsymbol{x}) \parallel \boldsymbol{v}\big)\big]}_{\text{output-space}} + \underbrace{\frac{\beta_k}{2}\|\boldsymbol{\theta} - \boldsymbol{\theta}_0\|_{\boldsymbol{\Lambda}_k}^2}_{\text{weight-space (Bayes/SSA proximal)}} \tag{4}$$

$\boldsymbol{p} = g_{\boldsymbol{\theta}}(\boldsymbol{x})$, i.e., the softmax probability output of the neural network on the test data and $\boldsymbol{q} = \boldsymbol{v}$, i.e., the uniform distribution on the class, $D_{\boldsymbol{\Phi}}(\boldsymbol{p}, \boldsymbol{q}) = \mathbb{KL}(g_{\boldsymbol{\theta}}(\boldsymbol{x}), \boldsymbol{v})$. We set $\boldsymbol{p} = \boldsymbol{\theta}$ and $\boldsymbol{q} = \boldsymbol{\theta}_*$, $D_{\boldsymbol{\Psi}}(\boldsymbol{p}, \boldsymbol{q}) = (\boldsymbol{\theta} - \boldsymbol{\theta}_*)^T \boldsymbol{\Lambda}_k (\boldsymbol{\theta} - \boldsymbol{\theta}_*)$

## 3.3 An Unlearning-Enhanced TTA Method

The general framework in section 3.2 reveals that existing TTA methods focus on the direct adaptation of the pre-trained model to the target test data distribution, but overlook an important fact that the distribution of test data often substantially diverges from that of the training data. Consequently, not

all the knowledge stored in a pre-trained model is beneficial for handling unseen test data. In fact, certain elements of this pre-trained model may impede the model's ability to adapt effectively.

To address this issue, we propose a new TTA paradigm built on our general framework. We introduce a novel unlearning-enhanced optimization principle for TTA, which for the first time formulates adaptation as a *two-sided process*: (i) selectively unlearning source-specific biases that hinder adaptation, and (ii) simultaneously adapting to new test data distribution. For illustration, we propose the following unlearning-enhanced TTA (*Lana*) to reflect this dual objective by integrating with Tent (Wang et al., 2021). It is important to note that integrating *Lana* with other TTA methods is straightforward. We thus omit the details. The learning objective is shown below:

$$\min_{\boldsymbol{\theta}} H(g_{\boldsymbol{\theta}+\boldsymbol{\delta}(\boldsymbol{\theta})}(\boldsymbol{x})), \tag{5}$$

$$\boldsymbol{\delta}(\boldsymbol{\theta}) := \arg\min_{\boldsymbol{\delta}} \alpha H(g_{\boldsymbol{\theta}+\boldsymbol{\delta}}(\boldsymbol{x})) - \mathcal{L}_{CE}(\mathcal{D}_{id}, \boldsymbol{\theta} + \boldsymbol{\delta}) \tag{6}$$

In Eq. (6), we (1) maximize the loss on the source domain data, i.e., $\mathcal{L}_{CE}(\mathcal{D}_{id}, \boldsymbol{\theta})$, to facilitate adaptive unlearning irrelevant source domain knowledge; and (2) minimize the entropy loss on the target test data distribution to ensure the unlearning on source domain does not degrade the performance on the target test data distribution. Importantly, our method *does not rely on* raw source data. It requires only the Fisher Information Matrix, which can be efficiently estimated. In Eq. (5), we optimize the test data entropy loss function initialized with the unlearning-enhanced model parameters to ensure adaptation to the test data distribution. In this regard, adaptive unlearning of certain source domain knowledge can be advantageous in facilitating adaptation to the target test data.

**Lana As a Special Case** In the following, we propose an algorithm to solve Eq. (5 and 6) and derive the algorithm from our general optimization framework in Eq. (2). We set $\beta = 0$. We then set $\boldsymbol{\Phi}$ to be the negative entropy function, i.e., $\boldsymbol{\Phi}(\boldsymbol{p}) = \sum_{i=1}^{i=n} \boldsymbol{p}_i \log \boldsymbol{p}_i$. We then set $\boldsymbol{p} = g_{\boldsymbol{\theta}}(\boldsymbol{x})$, i.e., the softmax probability output of the neural network on the test data and $\boldsymbol{q} = \boldsymbol{v}$, i.e., the uniform distribution on the class probability distribution. Then, we optimize Eq. (6) and (5) alternatively by gradient descent. In Eq. (6), we adopt first-order Taylor expansion on the first loss term as following:

$$H(g_{\boldsymbol{\theta}}(\boldsymbol{x})) \approx H(g_{\boldsymbol{\theta}_*}(\boldsymbol{x})) + \nabla_{\boldsymbol{\theta}} H(g_{\boldsymbol{\theta}_*}(\boldsymbol{x}))(\boldsymbol{\theta} - \boldsymbol{\theta}_*) \tag{7}$$

The second loss term in Eq. (6) is the cross entropy loss on the source domain data, which are unavailable during TTA. We adopt Taylor expansion to approximate it as the following:

$$\mathcal{L}_{CE}(\mathcal{D}_{id}, \boldsymbol{\theta}) \approx \mathcal{L}_{CE}(\mathcal{D}_{id}, \boldsymbol{\theta}_*) + \nabla_{\boldsymbol{\theta}} \mathcal{L}_{CE}(\mathcal{D}_{id}, \boldsymbol{\theta}_*)(\boldsymbol{\theta} - \boldsymbol{\theta}_*) + \frac{1}{2}(\boldsymbol{\theta} - \boldsymbol{\theta}_*)^T F(\boldsymbol{\theta} - \boldsymbol{\theta}_*) \tag{8}$$

where $F$ is the Fisher Information Matrix (FIM) of the loss $\mathcal{L}(\mathcal{D}_{id}, \boldsymbol{\theta})$ on the source domain data. Since $\nabla_{\boldsymbol{\theta}} \mathcal{L}_{CE}(\mathcal{D}_{id}, \boldsymbol{\theta}_*)$ is close to zero at the stationary point, i.e., $\boldsymbol{\theta}_*$, we thus only need to optimize the quadratic term in Eq. (8). In summary, the approximate loss for Eq. (6) can be expressed as:

$$\mathcal{L}_{unlearn} \approx \alpha \nabla_{\boldsymbol{\theta}} H(g_{\boldsymbol{\theta}_*}(\boldsymbol{x}))(\boldsymbol{\theta} - \boldsymbol{\theta}_*) - \frac{1}{2}(\boldsymbol{\theta} - \boldsymbol{\theta}_*)^T F(\boldsymbol{\theta} - \boldsymbol{\theta}_*) \tag{9}$$

We then take the gradient with respect to $\boldsymbol{\theta}$ for the right hand side of the Eq. (9), we can obtain:

$$\alpha \nabla_{\boldsymbol{\theta}} H(g_{\boldsymbol{\theta}_*}(\boldsymbol{x})) - F(\boldsymbol{\theta} - \boldsymbol{\theta}_*) = 0 \tag{10}$$

Solving the above equation leads to the following unlearning for the source domain model:

$$\boldsymbol{\theta}_f = \boldsymbol{\theta}_* + \alpha F^{-1} \nabla_{\boldsymbol{\theta}} H(g_{\boldsymbol{\theta}_*}(\boldsymbol{x})) \tag{11}$$

where in Eq. (11), the precondition matrix FIM $F^{-1}$ facilitates adaptive unlearning in source-domain pre-trained data knowledge. Its role is to enable a slower update of crucial parameters associated with the source domain to preserve important source domain knowledge, while permitting less critical parameters to undergo more rapid unlearning of irrelevant knowledge. This is because the FIM $F^{-1}$ indicates the parameter importance for source domain knowledge. FIM can be efficiently computed once before TTA using a small subset of unlabeled in-domain test examples (Niu et al., 2022). It's crucial to recognize that the Hessian matrix of the KL divergence aligns with the FIM, representing the local curvature of parameter changes. In practical terms, this relationship is denoted as $\nabla_{\boldsymbol{\delta}}^2 \mathbb{KL}(g_{\boldsymbol{\theta}}(\boldsymbol{x})||g_{\boldsymbol{\theta}+\boldsymbol{\delta}}(\boldsymbol{x}))|_{\boldsymbol{\delta}=\mathbf{0}} = F$ (Lemma 1 in Appendix C). This equation identifies the steepest direction for achieving the most rapid unlearning of the output probability distribution in the source domain. It is clear that *Lana*, seeks uniform low loss within a Riemannian manifold where each point

represents a probability distribution from the perspective of information geometry. This captures the underlying geometry in the model parameter space. This optimization objective promotes the neural network parameter whose entire neighborhoods in Riemannian manifold (characterized by $\mathbb{E}_{\boldsymbol{x}}\mathbb{KL}(g_{\boldsymbol{\theta}}(\boldsymbol{x})||g_{\boldsymbol{\theta}+\boldsymbol{\delta}}(\boldsymbol{x})) \leq \rho$) have uniformly low loss value. This is in contrast to existing TTA methods which treat all parameters in the Euclidean space in the same way, which may not be suitable for direct adaptation of the source domain model to the target domain. (More details and analysis can be found in Appendix C.) In practice, this loss landscape area often shows significantly improved generalization (Izmailov et al., 2018). To simplify computation, we employ a diagonal approximation of the FIM. The parameter $\alpha$ represents the unlearning rate. Since source domain data is not available during TTA, we follow (Niu et al., 2022) to efficiently estimate the FIM, which is only needed to be calculated once. The entire algorithm is shown in Algorithm 1 in Appendix. We also provide detailed theoretical analysis and proof in Appendix.

## 4 EXPERIMENT

### 4.1 SETUP

In this section, we perform experiments to evaluate the effectiveness of the proposed TTA method, *Lana*, compared to various existing methods. Specifically, we evaluate different methods with different model architectures under different normalization layers (including Batch Normalization (BN) (Ioffe & Szegedy, 2015), Group normalization (GN) (Wu & He, 2018) and Layer normalization (LN) (Ba et al., 2016)). We perform experiments on ImageNet-C (Deng et al., 2009; Hendrycks & Dietterich, 2019; Hendrycks et al., 2020) and DomainNet (Peng et al., 2019). These datasets are widely recognized and extensively utilized for assessing out-of-distribution generalization. ImageNet-C comprises a diverse set of challenges, including 15 different types of corrupted images falling into four primary categories: noise, blur, weather, and digital artifacts. DomainNet is a large-scale multi-source domain adaptation dataset. Following (Saito et al., 2019), we use a subset of DomainNet with 126 classes which consists of four domains (Clipart, Painting, Real and Sketch) with natural shift, known as DomainNet-126. In our study, we conduct a comparative analysis of our proposed method against the SOTA techniques. We conduct adaptation on ResNet-50-BN (R-50-BN), ResNet-50-GN (R-50-GN) (He et al., 2016) and VitBase-LN (Vit-LN) (Dosovitskiy et al., 2021). For experiments on DomainNet-126, we follow the architecture in (Liang et al., 2021).

**Baselines** Following (Niu et al., 2023), we compare to the following SOTA baselines, including Tent (Wang et al., 2021), EATA (Niu et al., 2022), AdaContrast (Chen et al., 2022a), SAR (Niu et al., 2023), DeYO Lee et al. (2024), TEA (Yuan et al., 2024).

**TTA scenarios** In alignment with the experimental setup described in (Niu et al., 2023), our study evaluates the performance of three distinct TTA scenarios. These scenarios encompass: (1) *Online Imbalanced Label Distribution Shifts*, where the imbalance ratio $r$ is calculated as $r = \frac{q_{max}}{q_{min}}$. Here, $q_{max}$ represents the proportion of the majority class within the dataset, while $q_{min}$ signifies the proportion of the minority class. As $r$ increases, existing TTA methods exhibit a decreasing level of performance. In accordance with (Niu et al., 2023), we set $r = \infty$, resulting in test samples being presented in class order. (2) *Mixed Distribution Shifts*, which involves evaluating different methods on a combination of 15 corruption types. (3) *Small Batch Size*, where it is observed that existing TTA methods' performance deteriorates as the batch size decreases. We conduct a comparative analysis between our method and existing ones, specifically assessing their performance when the batch size is set to 1.

**Implementation Details** Following (Niu et al., 2023), our experiments are conducted on ResNet50-BN, ResNet50-GN and VitBase-LN (Dosovitskiy et al., 2021), obtained from torchvision or timm. We employ SGD as the optimizer with momentum of 0.9. The batch size is set to be 64 (except for experiments with batch size=1). The number of adaptive unlearning step is set to be 1, i.e., $J = 1$, for efficiency. The learning rate is set to be 0.00025 for ResNet models and 0.001 for Vision Transformer models. We follow a similar test-time sample selection strategy as in (Niu et al., 2023), where samples with low loss values are chosen for calculating the TTA loss. For hyperparameter search, following (Yu et al., 2023), we use the first task of each dataset as the validation dataset to apply grid search for selecting the best hyperparameters. For example, we apply grid search that achieves the best TTA performance by adapting from uncorrupted ImageNet to Gaussian noise

Table 2: Comparisons with SOTA on ImageNet-C (severity level 5) by test accuracy (%) under **online imbalanced label shifts** (imbalance ratio = ∞). "BN"/"GN"/"LN" denote Batch/Group/Layer normalization.

| Method | Noise | | | Blur | | | | Weather | | | | Digital | | | | Avg |
|---|---|---|---|---|---|---|---|---|---|---|---|---|---|---|---|---|
| | Gauss | Shot | Impulse | Defocus | Glass | Motion | Zoom | Snow | Frost | Fog | Bright | Contrast | Elastic | Pixel | JPEG | |
| ResNet50 (BN) | 2.2 | 2.9 | 1.8 | 17.8 | 9.8 | 14.5 | 22.5 | 16.8 | 23.4 | 24.6 | 59.0 | 5.5 | 17.1 | 20.7 | 31.6 | 18.0 |
| • EATA | 0.3 | 0.3 | 0.3 | 0.2 | 0.2 | 0.5 | 0.9 | 0.8 | 0.9 | 1.8 | 3.5 | 0.2 | 0.8 | 1.2 | 0.9 | 0.9 |
| • AdaContrast | 0.1 | 0.1 | 0.1 | 0.9 | 0.8 | 1.3 | 2.2 | 0.6 | 0.4 | 2.6 | 3.0 | 8.8 | 0.9 | 1.2 | 0.8 | 1.6 |
| • SAR | 1.4 | 1.9 | 1.5 | 1.0 | 1.0 | 1.5 | 2.9 | 1.9 | 2.0 | 4.4 | 5.7 | 0.6 | 3.3 | 4.0 | 3.8 | 2.5 |
| • Tent | 1.2 | 1.4 | 1.4 | 1.0 | 0.9 | 1.2 | 2.6 | 1.7 | 1.8 | 3.6 | 5.0 | 0.5 | 2.6 | 3.2 | 3.1 | 2.1 |
| • Tent+Lana (Ours) | $2.1_{\pm0.3}$ | $1.8_{\pm0.1}$ | $2.6_{\pm0.3}$ | $1.2_{\pm0.1}$ | $1.2_{\pm0.1}$ | $1.8_{\pm0.1}$ | $3.2_{\pm0.2}$ | $2.2_{\pm0.2}$ | $3.1_{\pm0.3}$ | $4.7_{\pm0.1}$ | $5.9_{\pm0.1}$ | $0.7_{\pm0.1}$ | $3.5_{\pm0.1}$ | $4.1_{\pm0.1}$ | $4.0_{\pm0.1}$ | $2.8_{\pm0.1}$ |
| ResNet50 (GN) | 17.9 | 19.9 | 17.9 | 19.7 | 11.3 | 21.3 | 24.9 | 40.4 | 47.4 | 33.6 | 69.2 | 36.3 | 18.7 | 28.4 | 52.2 | 30.6 |
| • EATA | 27.0 | 28.3 | 28.1 | 14.9 | 17.1 | 24.4 | 25.3 | 32.2 | 32.0 | 39.8 | 66.7 | 33.6 | 24.5 | 41.9 | 38.4 | 31.6 |
| • AdaContrast | 0.1 | 0.1 | 0.1 | 0.5 | 0.7 | 0.3 | 0.2 | 0.3 | 0.5 | 0.3 | 0.2 | 0.6 | 0.3 | 0.2 | 0.2 | 0.3 |
| • SAR | 33.1 | 36.5 | 35.5 | 19.2 | 19.5 | 33.3 | 27.7 | 23.9 | 45.3 | 50.1 | 71.9 | 46.7 | 7.1 | 52.1 | 56.3 | 37.2 |
| • Tent | 2.6 | 3.3 | 2.7 | 13.9 | 7.9 | 19.5 | 17.0 | 16.5 | 21.9 | 1.8 | 70.5 | 42.2 | 6.6 | 49.4 | 53.7 | 22.0 |
| • Tent+Lana (Ours) | $35.3_{\pm1.2}$ | $35.9_{\pm0.5}$ | $35.6_{\pm1.0}$ | $18.8_{\pm0.5}$ | $19.2_{\pm1.0}$ | $33.5_{\pm0.7}$ | $23.9_{\pm3.5}$ | $33.1_{\pm4.6}$ | $45.0_{\pm0.4}$ | $48.2_{\pm1.2}$ | $71.6_{\pm0.2}$ | $45.7_{\pm0.2}$ | $8.5_{\pm1.6}$ | $51.3_{\pm0.7}$ | $56.6_{\pm1.0}$ | $37.4_{\pm0.8}$ |
| VitBase (LN) | 9.4 | 6.7 | 8.3 | 29.1 | 23.4 | 34.0 | 27.0 | 15.8 | 26.3 | 47.4 | 54.7 | 43.9 | 30.5 | 44.5 | 47.6 | 29.9 |
| • EATA | 35.9 | 34.6 | 36.7 | 45.3 | 47.2 | 49.3 | 47.7 | 56.5 | 55.4 | 62.2 | 72.2 | 21.7 | 56.2 | 64.7 | 63.7 | 49.9 |
| • AdaContrast | 0.1 | 0.1 | 0.1 | 4.4 | 5.0 | 6.5 | 8.5 | 1.8 | 2.3 | 13.4 | 17.1 | 32.5 | 3.8 | 6.4 | 3.1 | 7.0 |
| • SAR | 46.5 | 43.1 | 48.9 | 55.3 | 54.3 | 58.9 | 54.8 | 53.6 | 46.2 | 69.7 | 76.2 | 66.2 | 60.9 | 69.6 | 66.6 | 58.0 |
| • Tent | 32.7 | 1.4 | 34.6 | 54.4 | 52.3 | 58.2 | 52.2 | 7.7 | 12.0 | 69.3 | 76.1 | 66.1 | 56.7 | 69.4 | 66.4 | 47.3 |
| • Tent+Lana (Ours) | $50.4_{\pm1.2}$ | $50.2_{\pm1.5}$ | $51.4_{\pm0.3}$ | $55.6_{\pm0.2}$ | $54.7_{\pm0.1}$ | $59.4_{\pm0.2}$ | $56.1_{\pm0.1}$ | $62.9_{\pm2.1}$ | $62.2_{\pm1.9}$ | $70.1_{\pm0.2}$ | $76.6_{\pm0.1}$ | $66.2_{\pm0.0}$ | $62.6_{\pm0.2}$ | $70.1_{\pm0.1}$ | $67.4_{\pm0.1}$ | $61.1_{\pm0.3}$ |
| • TEA | 46.9 | 43.7 | 49.3 | 55.4 | 54.4 | 59.1 | 55.4 | 53.5 | 46.3 | 70.0 | 76.8 | 66.8 | 61.1 | 69.8 | 66.8 | 58.3 |
| • TEA+Lana (Ours) | $51.5_{\pm1.3}$ | $51.1_{\pm1.5}$ | $51.9_{\pm0.4}$ | $56.8_{\pm0.2}$ | $54.9_{\pm0.1}$ | $59.6_{\pm0.3}$ | $56.8_{\pm0.1}$ | $63.7_{\pm2.1}$ | $63.4_{\pm2.0}$ | $70.6_{\pm0.2}$ | $77.5_{\pm0.1}$ | $66.6_{\pm0.1}$ | $62.7_{\pm0.2}$ | $70.2_{\pm0.1}$ | $67.6_{\pm0.1}$ | $61.7_{\pm0.2}$ |
| • DeYO | 53.5 | 36.0 | 54.6 | 57.6 | 58.7 | 63.7 | 46.2 | 67.6 | 66.0 | 73.2 | 77.9 | 66.7 | 69.0 | 73.5 | 70.3 | 62.3 |
| • DeYO+Lana (Ours) | $55.0_{\pm0.6}$ | $35.7_{\pm25.2}$ | $56.1_{\pm0.7}$ | $59.9_{\pm0.3}$ | $58.5_{\pm0.2}$ | $65.6_{\pm0.2}$ | $46.6_{\pm15.2}$ | $68.8_{\pm0.2}$ | $68.0_{\pm0.2}$ | $74.0_{\pm0.1}$ | $80.1_{\pm0.2}$ | $67.8_{\pm0.1}$ | $70.0_{\pm0.1}$ | $74.8_{\pm0.2}$ | $71.6_{\pm0.3}$ | $63.5_{\pm1.5}$ |

Table 3: Comparisons with SOTA on ImageNet-C (severity level 5) by test accuracy (%) under **Batch Size = 1**. "BN"/"GN"/"LN" denote the Batch/Group/Layer normalization.

| Method | Noise | | | Blur | | | | Weather | | | | Digital | | | | Avg |
|---|---|---|---|---|---|---|---|---|---|---|---|---|---|---|---|---|
| | Gauss | Shot | Impulse | Defocus | Glass | Motion | Zoom | Snow | Frost | Fog | Bright | Contrast | Elastic | Pixel | JPEG | |
| ResNet50 (BN) | 2.2 | 2.9 | 1.9 | 17.9 | 9.8 | 14.8 | 22.5 | 16.9 | 23.3 | 24.4 | 58.9 | 5.4 | 17.0 | 20.6 | 31.6 | 18.0 |
| • EATA | 0.1 | 0.1 | 0.1 | 0.1 | 0.1 | 0.1 | 0.2 | 0.2 | 0.2 | 0.1 | 0.2 | 0.1 | 0.1 | 0.2 | 0.1 | 0.1 |
| • AdaContrast | 0.1 | 0.1 | 0.1 | 0.4 | 0.4 | 0.5 | 1.0 | 0.3 | 0.2 | 0.8 | 1.3 | 1.2 | 0.3 | 0.5 | 0.3 | 0.5 |
| • SAR | 0.1 | 0.1 | 0.1 | 0.1 | 0.1 | 0.1 | 0.1 | 0.1 | 0.1 | 0.1 | 0.1 | 0.1 | 0.1 | 0.1 | 0.1 | 0.1 |
| • Tent | 0.1 | 0.1 | 0.1 | 0.1 | 0.1 | 0.1 | 0.2 | 0.2 | 0.2 | 0.2 | 0.2 | 0.1 | 0.1 | 0.2 | 0.1 | 0.1 |
| • Tent+Lana (Ours) | 0.1 | 0.1 | 0.1 | 0.1 | 0.1 | 0.2 | 0.1 | 0.1 | 0.2 | 0.1 | 0.2 | 0.1 | 0.1 | 0.2 | 0.1 | 0.12 |
| ResNet50 (GN) | 18.0 | 19.8 | 17.9 | 19.8 | 11.4 | 21.4 | 24.9 | 40.4 | 47.3 | 33.6 | 69.3 | 36.3 | 18.6 | 28.4 | 52.3 | 30.6 |
| • EATA | 24.8 | 28.3 | 25.7 | 18.1 | 17.3 | 28.5 | 29.3 | 44.5 | 44.3 | 41.6 | 70.9 | 44.6 | 27.0 | 46.8 | 55.7 | 36.5 |
| • AdaContrast | 0.1 | 0.1 | 0.2 | 0.1 | 0.1 | 0.1 | 0.1 | 0.1 | 0.1 | 0.1 | 0.1 | 0.1 | 0.1 | 0.1 | 0.1 | 0.1 |
| • SAR | 23.4 | 26.6 | 23.9 | 18.4 | 15.4 | 28.6 | 30.4 | 44.9 | 44.7 | 25.7 | 72.3 | 44.5 | 14.8 | 47.0 | 56.1 | 34.5 |
| • Tent | 2.5 | 2.9 | 2.5 | 13.5 | 3.6 | 18.6 | 17.6 | 15.3 | 23.0 | 1.4 | 70.4 | 42.2 | 6.2 | 49.2 | 53.8 | 21.5 |
| • Tent+Lana (Ours) | $33.0_{\pm0.5}$ | $36.0_{\pm0.6}$ | $33.8_{\pm0.2}$ | $18.8_{\pm0.2}$ | $19.2_{\pm0.3}$ | $31.7_{\pm0.3}$ | $34.3_{\pm0.4}$ | $37.3_{\pm0.2}$ | $46.3_{\pm0.2}$ | $10.7_{\pm0.2}$ | $72.5_{\pm0.1}$ | $46.9_{\pm0.1}$ | $9.9_{\pm0.1}$ | $51.6_{\pm0.3}$ | $56.8_{\pm0.2}$ | $35.9_{\pm0.2}$ |
| VitBase (LN) | 9.5 | 6.7 | 8.2 | 29.0 | 23.4 | 33.9 | 27.1 | 15.9 | 26.5 | 47.2 | 54.7 | 44.1 | 30.5 | 44.5 | 47.8 | 29.9 |
| • EATA | 29.7 | 25.1 | 34.6 | 44.7 | 39.2 | 48.3 | 42.4 | 37.5 | 45.9 | 60.0 | 65.9 | 61.2 | 46.4 | 58.2 | 59.6 | 46.6 |
| • AdaContrast | 0.1 | 0.1 | 0.1 | 0.1 | 0.1 | 0.1 | 0.1 | 0.1 | 0.1 | 0.1 | 0.1 | 0.1 | 0.1 | 0.1 | 0.1 | 0.1 |
| • SAR | 40.8 | 36.4 | 41.5 | 53.7 | 50.7 | 57.5 | 52.8 | 59.1 | 50.7 | 68.1 | 74.6 | 65.7 | 57.9 | 68.9 | 65.9 | 56.3 |
| • Tent | 42.2 | 1.0 | 43.3 | 52.4 | 48.2 | 55.5 | 50.5 | 16.5 | 16.9 | 66.4 | 74.9 | 64.7 | 51.6 | 67.0 | 64.3 | 47.7 |
| • Tent+Lana (Ours) | $49.0_{\pm0.1}$ | $47.6_{\pm0.2}$ | $49.6_{\pm0.1}$ | $55.3_{\pm0.1}$ | $53.1_{\pm0.2}$ | $59.2_{\pm0.2}$ | $55.4_{\pm0.3}$ | $60.3_{\pm0.2}$ | $51.1_{\pm0.3}$ | $70.3_{\pm0.2}$ | $76.7_{\pm0.4}$ | $66.7_{\pm0.0}$ | $61.1_{\pm0.2}$ | $70.2_{\pm0.1}$ | $67.6_{\pm0.0}$ | $59.6_{\pm0.2}$ |
| • Tea | 43.1 | 37.3 | 43.4 | 54.5 | 51.7 | 59.1 | 54.6 | 59.4 | 51.2 | 70.3 | 75.2 | 66.2 | 59.3 | 71.0 | 67.1 | 57.6 |
| • Tea+Lana | $44.9_{\pm0.3}$ | $38.9_{\pm0.1}$ | $44.0_{\pm0.1}$ | $56.9_{\pm0.4}$ | $52.8_{\pm0.1}$ | $60.8_{\pm0.2}$ | $55.4_{\pm0.3}$ | $61.2_{\pm0.4}$ | $51.0_{\pm0.1}$ | $72.4_{\pm0.5}$ | $75.5_{\pm0.1}$ | $66.0_{\pm0.1}$ | $60.3_{\pm0.5}$ | $73.0_{\pm0.5}$ | $67.3_{\pm0.1}$ | $58.7_{\pm0.2}$ |
| • DeYO | 54.0 | 52.1 | 55.1 | 58.8 | 59.5 | 64.2 | 53.5 | 68.2 | 66.4 | 73.7 | 78.3 | 68.2 | 68.9 | 73.8 | 70.8 | 64.4 |
| • DeYO+Lana | $56.0_{\pm0.5}$ | $54.6_{\pm3.1}$ | $54.9_{\pm0.6}$ | $59.1_{\pm0.2}$ | $61.5_{\pm0.2}$ | $63.9_{\pm0.2}$ | $55.5_{\pm4.1}$ | $69.5_{\pm0.1}$ | $68.8_{\pm0.1}$ | $74.1_{\pm0.2}$ | $79.9_{\pm0.1}$ | $69.4_{\pm0.2}$ | $69.2_{\pm0.1}$ | $74.7_{\pm0.0}$ | $72.4_{\pm0.2}$ | $65.5_{\pm0.6}$ |

corrupted ImageNet. For DomainNet, we apply TTA by adapting from Clipart to Painting to select the optimal hyperparameters. During TTA, for trainable parameters of our method, we follow the approach presented in Tent (Wang et al., 2021) by adapting the affine parameters of group/layer normalization layers in ResNet50-GN/VitBase-LN. The method is evaluated over three runs, and results are presented as mean and standard deviation. We perform all the experiments on a single A6000 Nvidia GPU. Code will be released upon acceptance.

4.2 RESULTS

**Online Imbalanced Label Distribution Shifts** We compare with state-of-the-art TTA methods in online imbalanced label distribution shifts in Table 2. The results show that our method can improve by more than 3% compared to SAR on ImageNet-C under this challenging online imbalanced label distribution shifts with VitBase-LN. This improvement is significant considering the challenging data corruptions and large number image classes in ImageNet-C dataset. We can also observe that the network with batch normalization (BN) does not perform well across different compared methods. This aligns with the findings of (Niu et al., 2023), which assert that BN poses a substantial impediment to TTA performance in wild test scenarios.

**Batch Size = 1** We evaluate the effectiveness of existing TTA methods with batch size = 1 in Table 3. The results show that our method can improve by more than 3% compared to SAR on ImageNet-C under this challenging setting with test batch size = 1 with VitBase-LN.

**Mixed Distribution Shifts** We evaluate the effectiveness of existing TTA methods in mixed distribution shifts in Table 8 in Appendix. The results show that our method can improve by 1% compared to SAR on ImageNet-C under the mixture of 15 data corruptions with VitBase-LN.

The performance enhancement seen with VitBase surpasses that of ResNet, primarily due to VitBase's propensity for overfitting to the training data (Chen et al., 2022b). The overfitting can lead to the memorization of a greater amount of irrelevant information, making the process of unlearning both more beneficial and essential. All these results highlight the advantages of our method, *Lana*, and emphasize the importance of integrating unlearning in TTA.

**Results on DomainNet-126** Results on DomainNet-126 are shown in Table 6 in Appendix. We can observe that integrating Lana with Tent/AdaContrast further improves TTA.

### 4.3 ABLATION STUDY

**Hyperparameter Analysis**, we evaluate the sensitivity of hyperparameters $\alpha$ and $J$ in Table 7 in Appendix. This shows that as the unlearning rate increases, TTA performance first increases and then declines due to heightened unlearning effects. When $\alpha = 0.0$, i.e., there is no unlearning, the performance drops significantly. With unlearning, our method improves the TTA performance by more than 13.8%. This indicates the necessity and beneficial effect of unlearning. Moreover, an increase in the number of unlearning steps initially leads to a slight performance improvement, followed by a subsequent decrease. To optimize efficiency, we choose one step of unlearning.

**Efficiency Evaluation** To compare the running efficiency of the proposed method with existing methods, we evaluate their running efficiency in Table 9 in Appendix. To further improve runtime efficiency, we apply the unlearning step once every two adaptation iterations instead of every step. With this improvement, our method increases computational cost by less than 53% compared to Tent, while achieving substantially higher performance and remaining significantly faster than SAR and AdaContrast. Additional optimizations—such as sparse FIM masking and parameter freezing—can further reduce overhead.

**Impact on Forgetting of Source Domain Performance** Following (Niu et al., 2022; Zhang et al., 2023), we evaluate the performance on source domain after adapting the model to target domain. We present the results in Table 4. These results demonstrate that, after adapting to the target domain, the accuracy on the source domain remains largely unaffected, indicating minimal forgetting.

Table 4: Impact on source domain test accuracy after adaptation on each corrupted dataset on ImageNet-C (severity level 5) by test accuracy (%) under **online imbalanced label shifts** (imbalance ratio = $\infty$). "BN"/"GN"/"LN" denote Batch/Group/Layer normalization, respectively.

| Method | Source Model | Noise | | | Blur | | | | Weather | | | | Digital | | | |
|---|---|---|---|---|---|---|---|---|---|---|---|---|---|---|---|---|
| | | Gauss | Shot | Impulse | Defocus | Glass | Motion | Zoom | Snow | Frost | Fog | Bright | Contrast | Elastic | Pixel | JPEG |
| • Tent | 78.01 | 78.79 | 79.10 | 79.16 | 81.06 | 79.74 | 80.99 | 80.51 | 59.91 | 30.63 | 80.39 | 80.96 | 80.49 | 79.22 | 81.64 | 81.81 |
| • EATA | 78.01 | 72.03 | 74.34 | 70.54 | 77.33 | 75.08 | 73.07 | 74.30 | 75.17 | 76.49 | 77.64 | 77.01 | 33.63 | 75.13 | 77.53 | 78.15 |
| • AdaContrast | 78.01 | 72.20 | 74.67 | 71.18 | 78.01 | 75.11 | 73.42 | 74.76 | 75.35 | 76.67 | 77.87 | 77.23 | 34.5 | 75.84 | 78.23 | 78.82 |
| • SAR | 78.01 | 79.37 | **79.66** | 79.74 | 81.36 | 80.24 | **81.26** | 81.05 | **80.43** | 75.09 | 80.55 | **81.13** | 80.57 | **79.48** | 81.91 | 81.99 |
| • Tent+Lana (Ours) | 78.01 | **79.46** | 79.42 | **79.86** | **81.72** | **80.48** | 81.15 | **81.07** | 79.72 | **78.81** | 80.7 | 80.89 | **80.94** | 79.46 | **82.11** | 82.02 |

**Effect of Batch Size in Batch Normalization for TTA performance**. To evaluate the effect of different batch size for the network with batch normalization, we perform an evaluation with different batch sizes, i.e., 32 and 64 in Table 5.

Table 5: Comparisons with SOTA on ImageNet-C (severity level 5) by test accuracy (%) under different batch size (bs) with batch normalization.

| Method | | Noise | | | Blur | | | | Weather | | | | Digital | | | | |
|---|---|---|---|---|---|---|---|---|---|---|---|---|---|---|---|---|---|
| | | Gauss | Shot | Impulse | Defocus | Glass | Motion | Zoom | Snow | Frost | Fog | Bright | Contrast | Elastic | Pixel | JPEG | Avg |
| bs = 64 | • SAR | 33.0 | 34.9 | 33.9 | 29.8 | 29.8 | 44.2 | 49.9 | 48.9 | 43.0 | 58.3 | 67.4 | **40.1** | 55.7 | 59.5 | 53.5 | 45.5 |
| | • Lana (Ours) | **34.6** | **36.3** | **36.2** | **31.5** | **30.4** | **45.1** | **51.0** | **50.0** | **44.0** | **59.4** | **68.2** | 39.3 | **56.8** | **60.0** | **55.1** | **46.5** |
| bs = 32 | • SAR | 31.2 | 30.8 | 34.0 | 28.6 | 28.1 | 44.3 | 50.0 | 49.6 | 42.9 | 57.6 | 66.6 | 27.3 | 55.7 | 58.6 | 53.5 | 43.9 |
| | • Lana (Ours) | **33.2** | **34.2** | **35.3** | **28.8** | **29.8** | **45.5** | **50.7** | **50.8** | **44.3** | **58.7** | **67.4** | **31.1** | **56.6** | **60.2** | **53.6** | **45.3** |

**Integrating with Other TTA Base Methods** To evaluate the effectiveness of the proposed method integrating with other base approaches, we present the results in Table 12 in Appendix.

**Evaluation of different methods under a standard TTA setting** To assess the effectiveness of various TTA approaches under the standard TTA setting, we follow the setup outlined in (Yuan et al., 2024). The results are presented in Table 11 in Appendix.

## 5 CONCLUSION

This paper proposes a general framework for TTA. Based on the framework, we derive a novel unlearning-enhanced TTA method from our framework to further enhance the TTA performance. Extensive theoretical analysis and experiments on various TTA scenarios show the effectiveness of the proposed method.

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

## A  APPENDIX

In this appendix, we first cast exist existing TTA methods as special cases. We then provide detailed theoretical analysis. Next, we further present implementation details. After that, we provide more experimental results.

## B  RECAST EXISTING TTA METHODS INTO OUR UNIFIED AND GENERAL FRAMEWORK

**Tent As A Special Case** Tent (Wang et al., 2021) is a entropy minimization-based method. Specifically, it minimizes the following loss function:

$$\mathcal{L}^{TTA} = H(g_{\boldsymbol{\theta}}(\boldsymbol{x})) \tag{12}$$

where $H(g_{\boldsymbol{\theta}}(\boldsymbol{x}))$ is the entropy function on the classifier class probabilities output. The above loss function can be equivalently expressed as the following:

$$\mathcal{L}^{TTA} = -\mathbb{KL}(g_{\boldsymbol{\theta}}(\boldsymbol{x}), \boldsymbol{v}) \tag{13}$$

where we use $\mathbb{KL}$ to denote the KL-divergence between two probability distributions. $\boldsymbol{v}$ is a uniform class distribution on the output classes. In this case, in Eq. (2), we set $\alpha = -1, \beta = 0$ and take $\boldsymbol{\Phi}$ to be the negative entropy function, i.e., $\boldsymbol{\Phi}(\boldsymbol{p}) = \sum_{i=1}^{i=n} \boldsymbol{p}_i \log \boldsymbol{p}_i$. We set $\boldsymbol{p} = g_{\boldsymbol{\theta}}(\boldsymbol{x})$, i.e., the softmax probability output of the neural network on the test data and $\boldsymbol{q} = \boldsymbol{v}$, i.e., the uniform distribution on the class. $D_{\boldsymbol{\Phi}}(\boldsymbol{p}, \boldsymbol{q}) = \mathbb{KL}(g_{\boldsymbol{\theta}}(\boldsymbol{x}), \boldsymbol{v})$. We recover the Tent.

**SAR As A Special Case** SAR (Niu et al., 2023) is a sharpness-aware optimization (Foret et al., 2021)-based TTA method. Specifically, it first perturbs the model parameters by maximizing the following loss function:

$$\mathcal{L}^{TTA} = -\mathbb{KL}(g_{\boldsymbol{\theta}}(\boldsymbol{x}), \boldsymbol{v}) \tag{14}$$

Then, it obtains the perturbation $\varepsilon(\boldsymbol{\theta}) = \rho \operatorname{sign}(\nabla_{\boldsymbol{\theta}} \mathcal{L}_{\boldsymbol{\theta}}^{TTA}) |\nabla_{\boldsymbol{\theta}} \mathcal{L}_{\boldsymbol{\theta}}^{TTA}| / ||\nabla_{\boldsymbol{\theta}} \mathcal{L}_{\boldsymbol{\theta}}^{TTA}||_2$ Next, it minimizes the perturbed loss function $\mathcal{L}_{\boldsymbol{\theta}+\varepsilon(\boldsymbol{\theta})}^{TTA} = -\mathbb{KL}(g_{\boldsymbol{\theta}+\varepsilon(\boldsymbol{\theta})}(\boldsymbol{x}), \boldsymbol{v})$. In this case, in Eq. (2), we set $\alpha = -1, \beta = 0$ and take $\boldsymbol{\Phi}$ to be the negative entropy function, i.e., $\boldsymbol{\Phi}(\boldsymbol{p}) = \sum_{i=1}^{i=n} \boldsymbol{p}_i \log \boldsymbol{p}_i$. We set $\boldsymbol{p} = g_{\boldsymbol{\theta}+\varepsilon(\boldsymbol{\theta})}(\boldsymbol{x})$, i.e., the softmax probability output of the neural network on the test data and $\boldsymbol{q} = \boldsymbol{v}$, i.e., the uniform distribution on the class probability distribution. $D_{\boldsymbol{\Phi}}(\boldsymbol{p}, \boldsymbol{q}) = \mathbb{KL}(g_{\boldsymbol{\theta}+\varepsilon(\boldsymbol{\theta})}(\boldsymbol{x}), \boldsymbol{v})$. We then recovered the SAR method.

**EATA As A Special Case** EATA (Niu et al., 2022) is a weight-regularization-based technique. It achieves this by imposing a penalty on weight updates using the Fisher Information Matrix (FIM), $F$. EATA can be expressed as:

$$\mathcal{L}^{TTA} = -\mathbb{KL}(g_{\boldsymbol{\theta}}(\boldsymbol{x}), \boldsymbol{v}) + \beta(\boldsymbol{\theta} - \boldsymbol{\theta}_*)^T F(\boldsymbol{\theta} - \boldsymbol{\theta}_*) \tag{15}$$

In Eq. (2), we set $\alpha = -1$ and take $\boldsymbol{\Psi}(\boldsymbol{\theta}) = \frac{1}{2}\boldsymbol{\theta}^T F\boldsymbol{\theta}$. We set $\boldsymbol{p} = \boldsymbol{\theta}$ and $\boldsymbol{q} = \boldsymbol{\theta}_*$. $D_{\boldsymbol{\Psi}}(\boldsymbol{p}, \boldsymbol{q}) = (\boldsymbol{\theta} - \boldsymbol{\theta}_*)^T F(\boldsymbol{\theta} - \boldsymbol{\theta}_*)$. Then, we recovered the EATA method.

**SWR As A Special Case** SWR (Choi et al., 2022) is a weight-regularization-based method. Specifically, it optimizes the following objective:

$$\mathcal{L}^{TTA} = H(g_{\boldsymbol{\theta}}(\boldsymbol{x})) - \boldsymbol{\Lambda} H(\hat{g}_{\boldsymbol{\theta}}(\boldsymbol{x})) + \beta \sum_{l=1}^{l=L} m_l ||\boldsymbol{\theta}^l - \boldsymbol{\theta}_*^l||^2 \tag{16}$$

where $\boldsymbol{\Lambda} > 0$ is a constant and $\hat{g}_{\boldsymbol{\theta}}(\boldsymbol{x})$ is the average predication probability within a mini-batch. SWR maximizes this entropy $H(\hat{g}_{\boldsymbol{\theta}}(\boldsymbol{x}))$ to encourage the output probability distributions not too confident. $m_l$ is a layer-wise parameter penalty constant, $L$ is the total number of layers and $\boldsymbol{\theta}^l$ are the network parameters in the $l^{th}$ layer. Equivalently, it can be expressed as the following objective:

$$\mathcal{L}^{TTA} = -\mathbb{KL}(g_{\boldsymbol{\theta}}(\boldsymbol{x}), \boldsymbol{v}) + \mathbb{KL}(\hat{g}_{\boldsymbol{\theta}}(\boldsymbol{x}), \boldsymbol{v}) + \beta(\boldsymbol{\theta} - \boldsymbol{\theta}_*)^T M(\boldsymbol{\theta} - \boldsymbol{\theta}_*) \tag{17}$$

where $M$ is a diagonal matrix, where $M = diag(m_1, m_1, \cdots, m_2, m_2, \cdots, m_L, m_L, \cdots)$. The diagonal elements in $M$ are layer-wise penalty constants which indicates how fast those layer

parameters should be updated. In this case, in Eq. (2), we set $\alpha = -1$ and take $\boldsymbol{\Psi}(\boldsymbol{\theta}) = \frac{1}{2}\boldsymbol{\theta}^T M \boldsymbol{\theta}$. We set $\boldsymbol{p} = \boldsymbol{\theta}$ and $\boldsymbol{q} = \boldsymbol{\theta}_*$. $D_{\boldsymbol{\Psi}}(\boldsymbol{p}, \boldsymbol{q}) = (\boldsymbol{\theta} - \boldsymbol{\theta}_*)^T M (\boldsymbol{\theta} - \boldsymbol{\theta}_*)$. Then, we recovered the SWR.

**AdaContrast As a Special Case** AdaContrast (Chen et al., 2022a) is a *hard* pseudo-labeling based method. We denote $\boldsymbol{y}$ as the one-hot vector for the pseudo label predicted by the nearest neighbours in a target domain data point. AdaContrast optimizes the following objective:

$$\mathcal{L}^{TTA} = \mathbb{KL}(g_{\boldsymbol{\theta}}(\boldsymbol{x}), \boldsymbol{y}) \tag{18}$$

In this case, in Eq. (2), we set $\alpha = 1, \beta = 0$. We take $\boldsymbol{\Phi}$ to be the negative entropy function, i.e., $\boldsymbol{\Phi}(\boldsymbol{p}) = \sum_{i=1}^{i=n} \boldsymbol{p}_i \log \boldsymbol{p}_i$. We set $\boldsymbol{p} = g_{\boldsymbol{\theta}}(\boldsymbol{x})$, i.e., the softmax probability output on the test data $\boldsymbol{x}$ and $\boldsymbol{q}$ to be the one-hot vector of the ground truth class distribution. Then, $D_{\boldsymbol{\Phi}}(\boldsymbol{p}, \boldsymbol{q}) = \mathbb{KL}(g_{\boldsymbol{\theta}}(\boldsymbol{x}), \boldsymbol{y})$. We recovered AdaContrast.

**TAST As a Special Case** TAST (Jang et al., 2023) is a *soft* pseudo-labeling based method. We denote $\hat{g}_{\boldsymbol{\theta}}(\boldsymbol{x})$ as the soft pseudo-label on test data $\boldsymbol{x}$ predicted by the nearest neighbours. TAST optimizes the following:

$$\mathcal{L}^{TTA} = \mathbb{KL}(g_{\boldsymbol{\theta}}(\boldsymbol{x}), \hat{g}_{\boldsymbol{\theta}}(\boldsymbol{x})) \tag{19}$$

In Eq. (2), we set $\alpha = 1, \beta = 0$. We take $\boldsymbol{\Phi}$ to be the negative entropy function, i.e., $\boldsymbol{\Phi}(\boldsymbol{p}) = \sum_{i=1}^{i=n} \boldsymbol{p}_i \log \boldsymbol{p}_i$. We set $\boldsymbol{p} = g_{\boldsymbol{\theta}}(\boldsymbol{x})$, i.e., the softmax probability output of the neural network on the test data and $\boldsymbol{q} = \hat{g}_{\boldsymbol{\theta}}(\boldsymbol{x})$. Then, $D_{\boldsymbol{\Phi}}(\boldsymbol{p}, \boldsymbol{q}) = \mathbb{KL}(g_{\boldsymbol{\theta}}(\boldsymbol{x}), \hat{g}_{\boldsymbol{\theta}}(\boldsymbol{x}))$. We recovered TAST.

The unified and general optimization objective for TTA is defined as the following:

$$\mathcal{L}^{TTA} = \alpha \underbrace{D_{\boldsymbol{\Phi}}(g_{\boldsymbol{\theta}}(\boldsymbol{x}), \boldsymbol{z})}_{\text{output space}} + \beta \underbrace{D_{\boldsymbol{\Psi}}(\boldsymbol{\theta}, \boldsymbol{\theta}_*)}_{\text{weight space}} \pm \underbrace{\mathcal{L}_{CE}(\mathcal{D}_{id}, \boldsymbol{\theta}_*)}_{\text{ID data (optional)}} \tag{20}$$

The following is the definition of Bregman divergence:

$$D_{\boldsymbol{\Phi}}(p, q) = \boldsymbol{\Phi}(p) - \boldsymbol{\Phi}(q) - \langle \nabla \boldsymbol{\Phi}(q), p - q \rangle \tag{21}$$

## B.1 TENT/SAR AS A SPECIAL CASE

In Eq. (20), we set $\alpha = -1, \beta = 0$ and take $\boldsymbol{\Phi}(p) = \sum_{i=1}^{i=n} p_i \log p_i$. Here, $p$ and $q$ are probability simplex, i.e., $\sum_{i=1}^{i=n} p_i = 1$ and $\sum_{i=1}^{i=n} q_i = 1$. Then, we plug $\boldsymbol{\Phi}(p)$ into Eq. (21). We can obtain the following equation:

$$D_{\boldsymbol{\Phi}}(p, q) = \sum_{i=1}^{i=n} p_i \log p_i - \sum_{i=1}^{i=n} q_i \log q_i - \langle \log(q) + 1, p - q \rangle \tag{22}$$

$$= \sum_{i=1}^{i=n} p_i \log p_i - \sum_{i=1}^{i=n} p_i \log q_i - \sum_{i=1}^{i=n} p_i + \sum_{i=1}^{i=n} q_i \tag{23}$$

$$= \sum_{i=1}^{i=n} p_i \log \frac{p_i}{q_i} \tag{24}$$

$$= -H(p) + H(p, q) \tag{25}$$

$$= \mathbb{KL}(p||q) \tag{26}$$

where $H(p)$ is the entropy for the probability distribution $p$. and $H(p, q)$ is the cross entropy between probability distributions $p$ and $q$.

When we take the probability distribution $p = g_{\boldsymbol{\theta}}(\boldsymbol{x})$, i.e., the TTA model output probability distribution over the classes, and $q = \boldsymbol{v}$, i.e., the uniform distribution over the underlying classes, $D_{\boldsymbol{\Phi}}(p, q) = \mathbb{KL}(g_{\boldsymbol{\theta}}(\boldsymbol{x}), \boldsymbol{v})$. This precisely recovers the Tent/SAR method.

## B.2 EATA As a Special Case

In Eq. (20), we set $\alpha = 0$, we take $\boldsymbol{\Psi}(\boldsymbol{\theta}) = \frac{1}{2}\boldsymbol{\theta}^T F \boldsymbol{\theta}$. We set $p = \boldsymbol{\theta}$ and $q = \boldsymbol{\theta}_*$. where $F$ is the diagonal Fisher information matrix.

$$D_{\boldsymbol{\Phi}}(p, q) = \boldsymbol{\Phi}(p) - \boldsymbol{\Phi}(q) - \langle \nabla \boldsymbol{\Phi}(q), p - q \rangle \tag{27}$$

$$= \frac{1}{2}\boldsymbol{\theta}^T F \boldsymbol{\theta} - \frac{1}{2}\boldsymbol{\theta}_*^T F \boldsymbol{\theta}_* - \langle \boldsymbol{\theta}_* F, \boldsymbol{\theta} - \boldsymbol{\theta}_* \rangle \tag{28}$$

$$= \frac{1}{2}\boldsymbol{\theta}^T F \boldsymbol{\theta} + \frac{1}{2}\boldsymbol{\theta}_*^T F \boldsymbol{\theta}_* - \langle \boldsymbol{\theta}_* F, \boldsymbol{\theta} \rangle \tag{29}$$

$$= \frac{1}{2}(\boldsymbol{\theta} - \boldsymbol{\theta}_*)^T F (\boldsymbol{\theta} - \boldsymbol{\theta}_*) \tag{30}$$

Then, we recover the EATA method.

## B.3 SWR As A Special Case

SWR (Choi et al., 2022) is a weight-regularization-based method. Specifically, it optimizes the following objective:

$$\mathcal{L}^{TTA} = H(g_{\boldsymbol{\theta}}(\boldsymbol{x})) - \boldsymbol{\Lambda} H(\hat{g}_{\boldsymbol{\theta}}(\boldsymbol{x})) + \beta \sum_{l=1}^{l=L} m_l ||\boldsymbol{\theta}^l - \boldsymbol{\theta}_*^l||^2 \tag{31}$$

where $\boldsymbol{\Lambda} > 0$ is a constant and $\hat{g}_{\boldsymbol{\theta}}(\boldsymbol{x})$ is the average predication probability within a mini-batch. SWR maximizes this entropy $H(\hat{g}_{\boldsymbol{\theta}}(\boldsymbol{x}))$ to encourage the output probability distributions not too confident. $m_l$ is a layer-wise parameter penalty constant, $L$ is the total number of layers and $\boldsymbol{\theta}^l$ are the network parameters in the $l^{th}$ layer. Equivalently, it can be expressed as the following optimization objective:

$$\mathcal{L}^{TTA} = -\mathbb{KL}(g_{\boldsymbol{\theta}}(\boldsymbol{x}), \boldsymbol{v}) + \mathbb{KL}(\hat{g}_{\boldsymbol{\theta}}(\boldsymbol{x}), \boldsymbol{v}) +$$
$$\beta(\boldsymbol{\theta} - \boldsymbol{\theta}_*)^T M (\boldsymbol{\theta} - \boldsymbol{\theta}_*) \tag{32}$$

where $M$ is a diagonal matrix, where

$$M = diag(m_1, m_1, \cdots, m_2, m_2, \cdots, m_L, m_L, \cdots) \tag{33}$$

The diagonal elements in $M$ are layer-wise penalty constants which indicates how fast those layer parameters should be updated. In this case, in Eq. (2), we set $\alpha = -1$ and take $\boldsymbol{\Psi}(\boldsymbol{\theta}) = \frac{1}{2}\boldsymbol{\theta}^T M \boldsymbol{\theta}$. We set $p = \boldsymbol{\theta}$ and $q = \boldsymbol{\theta}_*$. By deriving the SWR as the following equation:

$$D_{\boldsymbol{\Phi}}(p, q) = \boldsymbol{\Phi}(p) - \boldsymbol{\Phi}(q) - \langle \nabla \boldsymbol{\Phi}(q), p - q \rangle \tag{34}$$

$$= \frac{1}{2}\boldsymbol{\theta}^T M \boldsymbol{\theta} - \frac{1}{2}\boldsymbol{\theta}_*^T M \boldsymbol{\theta}_* - \langle \boldsymbol{\theta}_* M, \boldsymbol{\theta} - \boldsymbol{\theta}_* \rangle \tag{35}$$

$$= \frac{1}{2}\boldsymbol{\theta}^T M \boldsymbol{\theta} + \frac{1}{2}\boldsymbol{\theta}_*^T M \boldsymbol{\theta}_* - \langle \boldsymbol{\theta}_* M, \boldsymbol{\theta} \rangle \tag{36}$$

$$= \frac{1}{2}(\boldsymbol{\theta} - \boldsymbol{\theta}_*)^T M (\boldsymbol{\theta} - \boldsymbol{\theta}_*) \tag{37}$$

Then, we recovered the SWR method.

## B.4 Fit LAME Loss into general TTA framework

$$\mathcal{L}^{\text{LAME}}(\tilde{Z}) = \sum_i \mathbb{KL}(\tilde{\boldsymbol{z}}_i \| \boldsymbol{q}_i) - \sum_{i,j} w_{ij} \tilde{\boldsymbol{z}}_i^\top \tilde{\boldsymbol{z}}_j$$

Where:

- $\tilde{z}_i$: soft pseudo-label for test input $x_i$, e.g., from $p_\theta(y|x_i)$
- $q_i$: smoothed or sharpened version of pseudo-label
- $w_{ij}$: similarity-based affinity weight (e.g., based on k-NN in feature space)

The KL divergence in the first term can be written as Bregman divergence as the following:

$$D_\phi(\tilde{z}_i, q_i) = \mathrm{KL}(\tilde{z}_i \parallel q_i)$$

We then formulate the second term: $\sum_{i,j} w_{ij} \tilde{z}_i^\top \tilde{z}_j$

as a Bregman divergence, we can reinterpret it through the lens of negative similarity minimization, which corresponds to Bregman divergence induced by a quadratic function.

If we choose:

$$\phi(z) = \frac{1}{2}\|z\|^2$$

then the corresponding Bregman divergence is:

$$D_\phi(z, z') = \frac{1}{2}\|z - z'\|^2$$

Note that:

$$\|z_i - z_j\|^2 = \|z_i\|^2 + \|z_j\|^2 - 2z_i^\top z_j$$

So:

$$z_i^\top z_j = \frac{1}{2}\left(\|z_i\|^2 + \|z_j\|^2 - \|z_i - z_j\|^2\right)$$

Therefore:

$$-z_i^\top z_j = -\frac{1}{2}\|z_i\|^2 - \frac{1}{2}\|z_j\|^2 + \frac{1}{2}\|z_i - z_j\|^2$$

Now summing over $i, j$ with weights $w_{ij}$, we get:

$$-\sum_{i,j} w_{ij} z_i^\top z_j = \sum_{i,j} w_{ij}\left[\frac{1}{2}\|z_i - z_j\|^2 - \frac{1}{2}\|z_i\|^2 - \frac{1}{2}\|z_j\|^2\right]$$

We can rewrite the *affinity regularization term* as a weighted sum of Bregman divergences:

$$\sum_{i,j} w_{ij} \cdot D_\phi(\tilde{z}_i, \tilde{z}_j) \quad \text{where} \quad D_\phi(\tilde{z}_i, \tilde{z}_j) = \frac{1}{2}\|\tilde{z}_i - \tilde{z}_j\|^2$$

Thus:

$$-\sum_{i,j} w_{ij} \tilde{z}_i^\top \tilde{z}_j \quad \longrightarrow \quad \sum_{i,j} w_{ij} D_\phi(\tilde{z}_i, \tilde{z}_j) + \text{const}$$

### B.5 FIT SSA LOSS INTO GENERAL TTA FRAMEWORK

**Objective.** We instantiate the general TTA objective as

$$J(\theta) = \underbrace{\alpha_k \, \mathbb{E}_{x \sim D_k}\big[\mathbb{KL}\big(p_\theta(\cdot \mid x) \parallel v\big)\big]}_{\text{output-space (entropy minimization)}} + \underbrace{\frac{\beta_k}{2}\|\theta - \theta_0\|_{\Lambda_k}^2}_{\text{weight-space (Bayes/SSA proximal)}} \tag{38}$$

where $v$ is the uniform distribution over classes, $\theta_0$ are the source weights, $\Lambda_k \simeq I$ (or a small diagonal), and $\beta_k \propto \sigma_\Lambda/\sigma_k^2$ with $\sigma_k^2$ an online estimate of gradient-noise variance.

**Forward (gradient) step on the output term.** Let

$$\ell(\boldsymbol{x}; \boldsymbol{\theta}) \;=\; -H\big(p_{\boldsymbol{\theta}}(\cdot \mid \boldsymbol{x})\big), \qquad \mathbf{g}_k \;=\; \nabla_{\boldsymbol{\theta}}\, \mathbb{E}_{x \sim D_k}\big[\ell(\boldsymbol{x}; \boldsymbol{\theta})\big]. \tag{39}$$

Take a covariance-aware scaled step

$$\mathbf{v}_k \;=\; \mathbf{m}_k \;-\; \alpha_k\, \eta\, \mathbf{g}_k, \tag{40}$$

with step scaler, e.g.,

$$\alpha_k \;=\; \sqrt{\frac{\sigma_\lambda^2}{\eta^2 \sigma_k^2}} \tag{41}$$

**Backward (proximal) step on the quadratic weight term.** We then solve the proximal subproblem

$$\boldsymbol{\theta}_{k+1} \;=\; \arg\min_{\boldsymbol{\theta}} \; \frac{1}{2\eta}\, \|\boldsymbol{\theta} - \mathbf{v}_k\|_2^2 \;+\; \frac{\beta_k}{2}\, \|\boldsymbol{\theta} - \boldsymbol{\theta}_0\|_{\boldsymbol{\Lambda}_k}^2. \tag{42}$$

Setting the gradient to zero gives

$$\frac{1}{\eta}(\boldsymbol{\theta} - \mathbf{v}_k) \;+\; \beta_k\, \boldsymbol{\Lambda}_k\, (\boldsymbol{\theta} - \boldsymbol{\theta}_0) \;=\; 0 \;\implies\; \big(I + \eta\beta_k\boldsymbol{\Lambda}_k\big)\boldsymbol{\theta} \;=\; \mathbf{v}_k \;+\; \eta\beta_k\boldsymbol{\Lambda}_k\,\boldsymbol{\theta}_0. \tag{43}$$

Hence

$$\boldsymbol{\theta}_{k+1} \;=\; \big(I + \eta\beta_k\boldsymbol{\Lambda}_k\big)^{-1}\Big(\mathbf{v}_k + \eta\beta_k\boldsymbol{\Lambda}_k\,\boldsymbol{\theta}_0\Big). \tag{44}$$

**Affine shrinkage (Kalman-style form).** Define the gain

$$A_k \;=\; \big(I + \eta\beta_k\boldsymbol{\Lambda}_k\big)^{-1}\eta\beta_k\boldsymbol{\Lambda}_k \;=\; I - \big(I + \eta\beta_k\boldsymbol{\Lambda}_k\big)^{-1}. \tag{45}$$

Then equation 44 is equivalently

$$\boldsymbol{\theta}_{k+1} \;=\; \mathbf{v}_k \;+\; A_k\,(\boldsymbol{\theta}_0 - \mathbf{v}_k). \tag{46}$$

Substituting equation 40 yields the Kalman-style mean update

$$\mathbf{m}_{k+1} \;=\; \big(\mathbf{m}_k - \alpha_k\, \eta\, \mathbf{g}_k\big) \;+\; A_k\Big(\boldsymbol{\theta}_0 - \big(\mathbf{m}_k - \alpha_k\, \eta\, \mathbf{g}_k\big)\Big). \tag{47}$$

**Parameter mapping and small-step limit.** If $\boldsymbol{\Lambda}_k \simeq I$ and $\eta\beta_k \ll 1$,

$$A_k \;\approx\; \eta\beta_k\, I \qquad \implies \qquad \beta_k \;\approx\; \frac{1}{\eta}\, a_k \quad \text{when } A_k = a_k I. \tag{48}$$

Together with $\beta_k \propto \sigma_\lambda / \sigma_k^2$ and the choice of $\alpha_k$ in equation 41, this recovers the Bayesian weight enhancement + steady-state adaptation behavior within our generalized TTA framework.

## C  THEORETICAL ANALYSIS

In this section, we perform theoretical analysis for our proposed method. In Theorem 2, *Lana* can be characterized as an optimization in a Riemannian manifold (defined as $\mathcal{M} = \{g_{\boldsymbol{\theta}}(\boldsymbol{x})\}$) to ensure uniform low loss in a probability distribution space. This helps achieve better generalization compared to achieve low loss in a single distribution. This is in contrast to existing TTA methods which treat all parameters in the Euclidean space in the same way without considering the underlying parameter geometry, which may not be suitable for direct adaptation of the source domain model to the target domain.

In Theorem C, we prove the generalization bound for *Lana*.

**Lemma 1**

$$\nabla_{\boldsymbol{\delta}}^2 \mathbb{KL}(p(\boldsymbol{x}|\boldsymbol{\theta})||p(\boldsymbol{x}|\boldsymbol{\theta}+\boldsymbol{\delta}))|_{\boldsymbol{\delta}=\mathbf{0}} = F \tag{49}$$

**proof**

First, the $\mathbb{KL}(p(\boldsymbol{x}|\boldsymbol{\theta})||p(\boldsymbol{x}|\boldsymbol{\theta}+\boldsymbol{\delta}))$ can be decomposed into the following:

$$
\begin{aligned}
\mathbb{KL}(p(\boldsymbol{x}|\boldsymbol{\theta})||p(\boldsymbol{x}|\boldsymbol{\theta}+\boldsymbol{\delta})) &= \mathbb{E}_{P(\boldsymbol{x}|\boldsymbol{\theta})}[\log P(\boldsymbol{x}|\boldsymbol{\theta})] \\
&- \mathbb{E}_{P(\boldsymbol{x}|\boldsymbol{\theta})}[\log P(\boldsymbol{x}|\boldsymbol{\theta}+\boldsymbol{\delta})]
\end{aligned}
\tag{50}
$$

$$
\begin{aligned}
\nabla_{\boldsymbol{\delta}}\mathbb{KL}(p(\boldsymbol{x}|\boldsymbol{\theta})||p(\boldsymbol{x}|\boldsymbol{\theta}+\boldsymbol{\delta})) &= \nabla_{\boldsymbol{\delta}}\mathbb{E}_{P(\boldsymbol{x}|\boldsymbol{\theta})}[\log P(\boldsymbol{x}|\boldsymbol{\theta})] \\
&- \nabla_{\boldsymbol{\delta}}\mathbb{E}_{P(\boldsymbol{x}|\boldsymbol{\theta})}[\log P(\boldsymbol{x}|\boldsymbol{\theta}+\boldsymbol{\delta})] \\
&= -\mathbb{E}_{P(\boldsymbol{x}|\boldsymbol{\theta})}[\nabla_{\boldsymbol{\delta}}\log P(\boldsymbol{x}|\boldsymbol{\theta}+\boldsymbol{\delta})] \\
&= -\int P(\boldsymbol{x}|\boldsymbol{\theta})\nabla_{\boldsymbol{\delta}}\log P(\boldsymbol{x}|\boldsymbol{\theta}+\boldsymbol{\delta})d\boldsymbol{x}
\end{aligned}
\tag{51}
$$

$$
\begin{aligned}
&\nabla_{\boldsymbol{\delta}}^2\mathbb{KL}(p(\boldsymbol{x}|\boldsymbol{\theta})||p(\boldsymbol{x}|\boldsymbol{\theta}+\boldsymbol{\delta})) \\
&= -\int P(\boldsymbol{x}|\boldsymbol{\theta})\nabla_{\boldsymbol{\delta}}^2\log P(\boldsymbol{x}|\boldsymbol{\theta}+\boldsymbol{\delta})d\boldsymbol{x}
\end{aligned}
\tag{52}
$$

Therefore, the Hessian with respect to $\boldsymbol{\delta}$ is as the following:

$$
\begin{aligned}
&\nabla_{\boldsymbol{\delta}}^2\mathbb{KL}(p(\boldsymbol{x}|\boldsymbol{\theta})||p(\boldsymbol{x}|\boldsymbol{\theta}+\boldsymbol{\delta}))|_{\boldsymbol{\delta}=\boldsymbol{0}} = \\
&= -\int P(\boldsymbol{x}|\boldsymbol{\theta})\nabla_{\boldsymbol{\delta}}^2\log P(\boldsymbol{x}|\boldsymbol{\theta}+\boldsymbol{\delta})|_{\boldsymbol{\delta}=\boldsymbol{0}}d\boldsymbol{x} \\
&= -\mathbb{E}_{P(\boldsymbol{x}|\boldsymbol{\theta})}Hessian(\log P(\boldsymbol{x}|\boldsymbol{\theta})) \\
&= F
\end{aligned}
\tag{53}
$$

Then, we apply Lemma 1 by setting $P(\boldsymbol{x}|\boldsymbol{\theta}+\boldsymbol{\delta}) = g_{\boldsymbol{\theta}+\boldsymbol{\delta}}(\boldsymbol{x})$ and $P(\boldsymbol{x}|\boldsymbol{\theta}) = g_{\boldsymbol{\theta}}(\boldsymbol{x})$. Then, we can obtain the conclusion that $\nabla_{\boldsymbol{\delta}}^2\mathbb{KL}(g_{\boldsymbol{\theta}}(\boldsymbol{x})||g_{\boldsymbol{\theta}+\boldsymbol{\delta}}(\boldsymbol{x}))|_{\boldsymbol{\delta}=\boldsymbol{0}} = F$

**Theorem 2** *With one step of adaptive unlearning by Eq. (11), Lana approximately minimizes the following flatness-seeking optimization objective. That is, solving Eq. (5) and Eq. (6) approximately solves the following optimization:*

$$
\min_{\boldsymbol{\theta}} \max_{\boldsymbol{\delta}:d(\boldsymbol{\theta},\boldsymbol{\theta}+\boldsymbol{\delta})\leq\sigma} H(g_{\boldsymbol{\theta}+\boldsymbol{\delta}}(\boldsymbol{x}))
\tag{54}
$$

$$
d(\boldsymbol{\theta},\boldsymbol{\theta}+\boldsymbol{\delta}) = \mathbb{E}_{\boldsymbol{x}}\mathbb{KL}(g_{\boldsymbol{\theta}}(\boldsymbol{x})||g_{\boldsymbol{\theta}+\boldsymbol{\delta}}(\boldsymbol{x}))
\tag{55}
$$

*where $\sigma > 0$ is a constant.*

**proof**

We take the first-order Taylor expansion on $H(g_{\boldsymbol{\theta}+\boldsymbol{\delta}}(\boldsymbol{x}))$ as the following:

$$
H(g_{\boldsymbol{\theta}+\boldsymbol{\delta}}(\boldsymbol{x})) \approx H(g_{\boldsymbol{\theta}}(\boldsymbol{x})) + \nabla_{\boldsymbol{\theta}}H(g_{\boldsymbol{\theta}}(\boldsymbol{x}))^T\boldsymbol{\delta}
\tag{56}
$$

According to Lemma 1, we use second-order Taylor expansion at $\boldsymbol{\delta}=\boldsymbol{0}$ as the following:

$$
\begin{aligned}
d(\boldsymbol{\theta},\boldsymbol{\theta}+\boldsymbol{\delta}) =& \mathbb{E}_{\boldsymbol{x}}[\underbrace{\mathbb{KL}(g_{\boldsymbol{\theta}}(\boldsymbol{x})||g_{\boldsymbol{\theta}+\boldsymbol{\delta}}(\boldsymbol{x}))|_{\boldsymbol{\delta}=\boldsymbol{0}}}_{=0} + \underbrace{\nabla_{\boldsymbol{\delta}}\mathbb{KL}(g_{\boldsymbol{\theta}}(\boldsymbol{x})||g_{\boldsymbol{\theta}+\boldsymbol{\delta}}(\boldsymbol{x}))|_{\boldsymbol{\delta}=\boldsymbol{0}}}_{=0}\boldsymbol{\delta} \\
&+ \frac{1}{2}\boldsymbol{\delta}^T\nabla_{\boldsymbol{\delta}}^2\underbrace{\mathbb{KL}(p(\boldsymbol{x}|\boldsymbol{\theta})||p(\boldsymbol{x}|\boldsymbol{\theta}+\boldsymbol{\delta}))|_{\boldsymbol{\delta}=\boldsymbol{0}}}_{=F \text{ by Lemma 1}}\boldsymbol{\delta}] + O(\boldsymbol{\delta}^3)
\end{aligned}
\tag{57}
$$

Therefore,

$$d(\boldsymbol{\theta}, \boldsymbol{\theta} + \boldsymbol{\delta}) \approx \frac{1}{2}\boldsymbol{\delta}^T F \boldsymbol{\delta} \tag{58}$$

Then, optimize the following optimization problem

$$\max_{d(\boldsymbol{\theta}, \boldsymbol{\theta}+\boldsymbol{\delta}) \leq \sigma} H(g_{\boldsymbol{\theta}+\boldsymbol{\delta}}(\boldsymbol{x})) \tag{59}$$

We can convert this constrained optimization by Lagrangian duality (Boyd & Vandenberghe, 2004), we can obtain the following:

$$\max_{\boldsymbol{\delta}} [H(g_{\boldsymbol{\theta}}(\boldsymbol{x})) + \nabla_{\boldsymbol{\theta}} H(g_{\boldsymbol{\theta}}(\boldsymbol{x}))^T \boldsymbol{\delta} - \gamma(\frac{1}{2}\boldsymbol{\delta}^T F \boldsymbol{\delta} - \sigma)] \tag{60}$$

We take the gradient with respect to $\boldsymbol{\delta}$ and obtain the following:

$$\nabla_{\boldsymbol{\theta}} H(g_{\boldsymbol{\theta}}(\boldsymbol{x})) - \gamma F \boldsymbol{\delta} = 0 \tag{61}$$

$$\boldsymbol{\delta} = \frac{1}{\gamma} F^{-1} \nabla_{\boldsymbol{\theta}} H(g_{\boldsymbol{\theta}}(\boldsymbol{x})) \tag{62}$$

This corresponds to the adaptive unlearning step. Then, the conclusion follows.

**Theorem** We denote the distribution of $\boldsymbol{x}$ as $\mathcal{D}$ and the test set $S$, which is sampled independently and identically distributed (i.i.d.) from $\mathcal{D}$. *Lana* Eq. (5) and Eq. (6) is equal to following optimization objective.

$$\min_{\boldsymbol{\theta}} H(g_{\boldsymbol{\theta}+\boldsymbol{\delta}(\boldsymbol{\theta})}(\boldsymbol{x})), \tag{63}$$

$$\boldsymbol{\delta}(\boldsymbol{\theta}) := \alpha F^{-1} \nabla_{\boldsymbol{\theta}} H(g_{\boldsymbol{\theta}}(\boldsymbol{x})) \tag{64}$$

And we have the generalization bound:

$$\mathbb{E}_{\boldsymbol{x} \sim \mathcal{D}}[H(g_{\boldsymbol{\theta}}(\boldsymbol{x}))] \leq \mathbb{E}_{\boldsymbol{x} \sim S}[H(g_{\boldsymbol{\theta}+\boldsymbol{\delta}(\boldsymbol{\theta})}(\boldsymbol{x}))] + (\frac{m^2}{d} e^{1-\frac{m^2}{d}})^{d/2} \tag{65}$$

$$+ \sqrt{\frac{\frac{1}{4}k \log\left(1 + \frac{m^2 \|\boldsymbol{\delta}\|_2^2}{k\sigma(\boldsymbol{\theta})^2}\right) + \frac{1}{4} + \log\frac{n}{\delta} + 2\log(6n + 3k)}{n-1}}, \tag{66}$$

where $d$ is the dimension of $\boldsymbol{\theta}$, $m$ is an arbitrary constant, and where $\sigma(\boldsymbol{\theta})$ is positively related to the scale of $\|\boldsymbol{\delta}(\boldsymbol{\theta})\|_2$.

By PAC-Bayesian bound (McAllester, 1999; Dziugaite & Roy, 2017) we have with probability at least $1 - \delta$ over the test set $S$, the following generalization bound holds for any prior $P$ and posterior $Q$ over parameters:

$$\mathbb{E}_{\boldsymbol{\theta} \sim Q}[\mathbb{E}_{\boldsymbol{x} \sim \mathcal{D}}[H(g_{\boldsymbol{\theta}}(\boldsymbol{x}))]] \leq \mathbb{E}_{\boldsymbol{\theta} \sim Q}[\mathbb{E}_{\boldsymbol{x} \sim S}[H(g_{\boldsymbol{\theta}}(\boldsymbol{x}))]] + \sqrt{\frac{KL(Q||P) + \log\frac{n}{\delta}}{2(n-1)}},$$

where $n = |S|$ and $k$ is the number of parameters. Besides, if $P = \mathcal{N}(\boldsymbol{\mu}_P, \sigma_P^2 \boldsymbol{I})$ and $Q = \mathcal{N}(\boldsymbol{\mu}_Q, \sigma_Q^2 \boldsymbol{I})$, then the KL divergence can be written as follows:

$$KL(Q||P) = \frac{1}{2}\left[\frac{k\sigma_Q^2 + \|\boldsymbol{\mu}_P - \boldsymbol{\mu}_Q\|_2^2}{\sigma_P^2} - k + k \log\left(\frac{\sigma_P^2}{\sigma_Q^2}\right)\right] \tag{67}$$

Following Theorem 1 of (Foret et al., 2021), we have with probability $1 - \frac{6\delta}{\pi^2 j^2}$, the KL-divergence is bounded by

$$KL(Q\|P) \leq \frac{1}{2}\left[1 + k\log\left(1 + \frac{\|\boldsymbol{\theta}\|_2^2}{k\sigma_Q^2}\right)\right], \tag{68}$$

where $j \leq \lfloor k\log(1 + \exp(4n/k))\rfloor$.

We assumed $\mathbb{E}_{\boldsymbol{x}\sim\mathcal{D}}[H(g_{\boldsymbol{\theta}}(\boldsymbol{x}))] \leq \mathbb{E}_{\boldsymbol{\delta}\sim\mathcal{N}(0,\sigma\boldsymbol{I})}[\mathbb{E}_{\boldsymbol{x}\sim\mathcal{D}}[H(g_{\boldsymbol{\theta}+\boldsymbol{\delta}}(\boldsymbol{x}))]]$, with the above derivation, the generalization bound can be written as follows:

$$\mathbb{E}_{\boldsymbol{x}\sim\mathcal{D}}[H(g_{\boldsymbol{\theta}}(\boldsymbol{x}))] \leq \mathbb{E}_{\boldsymbol{\delta}\sim\mathcal{N}(0,\sigma\boldsymbol{I})}[\mathbb{E}_{\boldsymbol{x}\sim S}[H(g_{\boldsymbol{\theta}+\boldsymbol{\delta}}(\boldsymbol{x}))]] \tag{69}$$

$$+ \sqrt{\frac{\frac{1}{4}k\log\left(1 + \frac{\|\boldsymbol{\delta}\|_2^2}{k\sigma^2}\right) + \frac{1}{4} + \log\frac{n}{\delta} + 2\log(6n + 3k)}{n - 1}} \tag{70}$$

Now we consider the relation between $\max_{\boldsymbol{\delta}:d(\boldsymbol{\theta},\boldsymbol{\theta}+\boldsymbol{\delta})\leq\rho} H(g_{\boldsymbol{\theta}+\boldsymbol{\delta}}(\boldsymbol{x}))$ and $\mathbb{E}_{\boldsymbol{\delta}\sim\mathcal{N}(0,\sigma\boldsymbol{I})}[\mathbb{E}_{\boldsymbol{x}\sim S}[H(g_{\boldsymbol{\theta}+\boldsymbol{\delta}}(\boldsymbol{x}))]]$. As $d(\boldsymbol{\theta},\boldsymbol{\theta}+\boldsymbol{\delta})$ is a continuous distance metric, there exist $\sigma(\boldsymbol{\theta})$ such that $\{\|\boldsymbol{\delta}\|_2 \leq \sigma(\boldsymbol{\theta})\} \subset \{\|d(\boldsymbol{\theta},\boldsymbol{\theta}+\boldsymbol{\delta})\|_2 \leq \rho\}$. In this case we have

$$\max_{\boldsymbol{\delta}:d(\boldsymbol{\theta},\boldsymbol{\theta}+\boldsymbol{\delta})\leq\rho} H(g_{\boldsymbol{\theta}+\boldsymbol{\delta}}(\boldsymbol{x})) \geq \max_{\|\boldsymbol{\delta}\|_2\leq\sigma(\boldsymbol{\theta})} \mathbb{E}_{\boldsymbol{x}\sim S}[H(g_{\boldsymbol{\theta}+\boldsymbol{\delta}}(\boldsymbol{x}))]$$

Consider $\boldsymbol{\delta} \sim \mathcal{N}(0, \sigma^2\boldsymbol{I})$, we have $\frac{\boldsymbol{\delta}}{\sigma} \sim \mathcal{N}(0, I_d)$ and $\forall m > 0$

$$\begin{aligned}
&\mathbb{E}_{\boldsymbol{\delta}\sim\mathcal{N}(0,\sigma\boldsymbol{I})}[\mathbb{E}_{\boldsymbol{x}\sim S}[H(g_{\boldsymbol{\theta}+\boldsymbol{\delta}}(\boldsymbol{x}))]]\\
&= \mathbb{E}_{\boldsymbol{\delta}}[\mathbb{E}_{\boldsymbol{x}\sim S}[H(g_{\boldsymbol{\theta}+\boldsymbol{\delta}}(\boldsymbol{x}))|\,\|\frac{\boldsymbol{\delta}}{\sigma}\|_2 \leq m]\mathbb{P}(\|\frac{\boldsymbol{\delta}}{\sigma}\|_2 \leq m)\\
&\quad+ \mathbb{E}_{\boldsymbol{\delta}}[\mathbb{E}_{\boldsymbol{x}\sim S}[H(g_{\boldsymbol{\theta}+\boldsymbol{\delta}}(\boldsymbol{x}))|\,\|\frac{\boldsymbol{\delta}}{\sigma}\|_2 > m]\mathbb{P}(\|\frac{\boldsymbol{\delta}}{\sigma}\|_2 > m))\\
&\leq \max_{\|\boldsymbol{\delta}\|_2\leq m\sigma}[\mathbb{E}_{\boldsymbol{x}\sim S}[H(g_{\boldsymbol{\theta}+\boldsymbol{\delta}}(\boldsymbol{x}))]]\mathbb{P}(\|\frac{\boldsymbol{\delta}}{\sigma}\|_2 \leq m) + \mathbb{P}(\|\frac{\boldsymbol{\delta}}{\sigma}\|_2 > m).\\
&\leq \max_{\|\boldsymbol{\delta}\|_2\leq m\sigma}[\mathbb{E}_{\boldsymbol{x}\sim S}[H(g_{\boldsymbol{\theta}+\boldsymbol{\delta}}(\boldsymbol{x}))] + \mathbb{P}(\|\frac{\boldsymbol{\delta}}{\sigma}\|_2 > m).
\end{aligned} \tag{71}$$

As $\frac{\boldsymbol{\delta}}{\sigma} \sim \mathcal{N}(0, \boldsymbol{I})$, by Chernoff bound of chi-squared distribution we have,

$$\mathbb{P}(\|\frac{\boldsymbol{\delta}}{\sigma}\|_2 > m) \leq \left(\frac{m^2}{d}e^{1-\frac{m^2}{d}}\right)^{d/2}, \tag{72}$$

where $d$ is the number of parameters. Thus by taking $\sigma = \sigma(\boldsymbol{\theta})/m$ we have

$$\mathbb{E}_{\boldsymbol{\delta}\sim\mathcal{N}(0,\sigma(\boldsymbol{\theta})/m\boldsymbol{I})}[\mathbb{E}_{\boldsymbol{x}\sim S}[H(g_{\boldsymbol{\theta}+\boldsymbol{\delta}}(\boldsymbol{x}))]] \leq \max_{\|\boldsymbol{\delta}\|_2\leq\sigma(\boldsymbol{\theta})}[\mathbb{E}_{\boldsymbol{x}\sim S}[H(g_{\boldsymbol{\theta}+\boldsymbol{\delta}}(\boldsymbol{x}))]] + \left(\frac{m^2}{d}e^{1-\frac{m^2}{d}}\right)^{d/2}, \tag{73}$$

Combining the above equation with Eq. (69), we have

$$
\begin{aligned}
\mathbb{E}_{\boldsymbol{x}\sim\mathcal{D}}[H(g_{\boldsymbol{\theta}}(\boldsymbol{x}))] &\leq \mathbb{E}_{\boldsymbol{\delta}\sim\mathcal{N}(0,\sigma(\boldsymbol{\theta})/m\boldsymbol{I})}[\mathbb{E}_{\boldsymbol{x}\sim S}[H(g_{\boldsymbol{\theta}+\boldsymbol{\delta}}(\boldsymbol{x}))]] \\
&\quad + \sqrt{\frac{\frac{1}{4}k\log\left(1+\frac{m^2\|\boldsymbol{\delta}\|_2^2}{k\sigma(\boldsymbol{\theta})^2}\right)+\frac{1}{4}+\log\frac{n}{\delta}+2\log\left(6n+3k\right)}{n-1}} \\
&\leq \max_{\|\boldsymbol{\delta}\|_2\leq\sigma(\boldsymbol{\theta})}[\mathbb{E}_{\boldsymbol{x}\sim S}[H(g_{\boldsymbol{\theta}+\boldsymbol{\delta}}(\boldsymbol{x}))]]+(\frac{m^2}{d}e^{1-\frac{m^2}{d}})^{d/2} \\
&\quad + \sqrt{\frac{\frac{1}{4}k\log\left(1+\frac{m^2\|\boldsymbol{\delta}\|_2^2}{k\sigma(\boldsymbol{\theta})^2}\right)+\frac{1}{4}+\log\frac{n}{\delta}+2\log\left(6n+3k\right)}{n-1}} \\
&\leq \max_{\boldsymbol{\delta}:d(\boldsymbol{\theta},\boldsymbol{\theta}+\boldsymbol{\delta})\leq\rho} H(g_{\boldsymbol{\theta}+\boldsymbol{\delta}}(\boldsymbol{x}))+(\frac{m^2}{d}e^{1-\frac{m^2}{d}})^{d/2} \\
&\quad + \sqrt{\frac{\frac{1}{4}k\log\left(1+\frac{m^2\|\boldsymbol{\delta}\|_2^2}{k\sigma(\boldsymbol{\theta})^2}\right)+\frac{1}{4}+\log\frac{n}{\delta}+2\log\left(6n+3k\right)}{n-1}}.
\end{aligned}
\tag{74}
$$

# D  MORE IMPLEMENTATION DETAILS

## D.1  DATASET DETAILS

- **ImageNet-C**: ImageNet-C is a dataset created for the purpose of evaluating the robustness and generalization ability of computer vision models. It is a corruption dataset, meaning that it contains images that have been corrupted to simulate real-world challenges that models might face. The corruption consists of 15 different types, i.e., Gaussian noise, shot noise, impulse noise, defocus blur, glass blue, motion blur, zoom blur, snow, frost, fog, brightness, contrast, elastic transformation, pixelation, and JPEG compression. Each corruption type further contains 5 different severity levels and the larger severity level means more severe distribution shift. ImageNet-C can assess how well a computer vision model trained on clean data can perform on images that have been corrupted in various ways. This helps in understanding the model's resilience to different types of distortions and aids in the development of more robust and generalizable models.

- **DomainNet**: DomainNet is a comprehensive multi-source domain adaptation dataset. In line with the methodology proposed by (Saito et al., 2019), we employ a specific subset of DomainNet called DomainNet-126, which comprises 126 classes distributed across four distinct domains: Clipart, Painting, Real, and Sketch. This subset is particularly notable for its representation of natural shifts inherent in real-world data.

## D.2  BASELINE DETAILS

- Tent (Wang et al., 2021): Tent is an entropy-minimization based TTA method. We follow the hyper-parameter setting in Tent (Wang et al., 2021). In particular, we employ SGD as the optimizer, incorporating a momentum factor of 0.9 and utilizing a batch size of 64. The chosen learning rates are 0.00025 for ResNet models and 0.001 for Vit models. Notably, when the batch size is 1, the learning rates are adjusted to 0.00025/32 for ResNet models and 0.001/64 for Vit models. The trainable parameters involves adjusting the affine parameters of normalization layers.

- EATA (Niu et al., 2022): EATA is a weight-regularization-based method. It regularizes the TTA model updates so that weights are important to the source-domain domain will be updated slower and weights that are less important to the source-domain will be updated slower.

- SAR (Niu et al., 2023): SAR, a sharpness-aware entropy minimization technique, enhances TTA stability by addressing two key issues: eliminating partially noisy samples characterized by large gradients and promoting the convergence of model weights towards a flat minimum. This approach ensures that the model becomes resilient to the presence of the remaining noisy samples.

- AdaContrast (Chen et al., 2022a) is an online pseudo labeling method combined with contrastive learning to perform TTA.

# E  MORE EXPERIMENTAL RESULTS

**Impact on Source Domain Performance** Following (Niu et al., 2022; Zhang et al., 2023), we also evaluate the performance on source domain after adapting the model to the target domain test data. We present the results in Table 4. We can observe that our method (Lana) overally outperforms existing TTA methods in terms of source-domain accuracy. This indicates that our method leads to minimal forgetting of source domain knowledge since our adaptive unlearning strategy considers the parameter importance with respect to the source domain and constrains the forgetting on source domain.

**Effect of Batch Size for Batch Normalization**. To evaluate the effect of different batch size for the network with batch normalization, we perform an evaluation with different batch sizes, i.e., 32 and 64 in Table 5. The results indicate that our method improves more than 1% compared to SOTA TTA method with different batch sizes.

**DomainNet-126 Results**

Table 6: Comparisons with SOTA on **DomainNet-126** by test accuracy (%). C, P, R and S denote the domain of Clipart, Painting, Real and Sketch, respectively. $\rightarrow$ indicates the transfer direction.

| Method | C→P | C→R | C→S | P→C | P→R | P→S | R→C | R→P | R→S | S→C | S→P | S→R | Avg |
|---|---|---|---|---|---|---|---|---|---|---|---|---|---|
| Source Model | 49.1 | 62.0 | 50.5 | 56.7 | 74.9 | 48.2 | 58.3 | 53.0 | 60.0 | 57.2 | 62.7 | 47.8 | 56.7 |
| • Tent | 53.1 | 64.1 | 53.1 | **57.3** | **73.9** | 56.8 | 58.0 | 65.6 | 52.0 | 63.1 | 62.0 | 66.6 | 60.5 |
| • Tent+Lana (Ours) | **54.5** | **66.4** | **54.7** | 57.2 | 73.3 | **57.0** | **58.5** | **66.7** | **52.1** | **63.9** | **63.8** | **67.8** | **61.3** |
| • EATA | 54.3 | 65.3 | 54.0 | 58.4 | 73.4 | 57.3 | 57.7 | 64.2 | 51.9 | 63.6 | 62.4 | 67.9 | 60.9 |
| • SAR | 53.0 | 64.8 | 53.1 | 56.8 | 73.7 | 56.7 | 56.7 | 64.3 | 51.3 | 62.9 | 62.2 | 67.5 | 60.2 |
| • AdaContrast | 57.2 | 69.6 | 56.3 | 61.8 | 77.2 | 60.3 | 62.2 | 66.3 | 54.3 | 67.2 | 65.2 | 72.2 | 64.1 |
| • AdaContrast + Lana (Ours) | **57.6**$_{\pm0.1}$ | **70.4**$_{\pm0.1}$ | **56.8**$_{\pm0.0}$ | **62.8**$_{\pm0.1}$ | **78.2**$_{\pm0.1}$ | **60.9**$_{\pm0.0}$ | **62.6**$_{\pm0.0}$ | **67.1**$_{\pm0.1}$ | **55.0**$_{\pm0.0}$ | **67.8**$_{\pm0.0}$ | **65.8**$_{\pm0.1}$ | **72.9**$_{\pm0.0}$ | **64.8**$_{\pm0.1}$ |

**Hyperparameter Sensitivity Analysis**

Table 7: Analysis of unlearning rate $\alpha$ and unlearning steps $J$ under online imbalanced label shifts with VitBase.

| $\alpha$ | 0.0 | 0.03 | 0.05 | 0.07 |
|---|---|---|---|---|
| Accuracy | 47.3 | 60.9 | 61.1 | 60.5 |

| $J$ | 1 | 2 | 3 |
|---|---|---|---|
| Accuracy | 61.1 | 61.3 | 60.4 |

**Mixture of 15 different corrupted data distributions**

Table 8: Comparisons with SOTA test accuracy on ImageNet-C under **mixture of 15 different corrupted data distributions**.

| Method | Accuracy | Method | Accuracy |
|---|---|---|---|
| ResNet50 (GN) | 30.6 | VitBase (LN) | 29.9 |
| • Tent | 13.4 | • Tent | 16.5 |
| • EATA | 38.1 | • EATA | 55.7 |
| • AdaContrast | 0.38 | • AdaContrast | 1.26 |
| • SAR | 38.3 | • SAR | 57.1 |
| • Tent+Lana (Ours) | **38.5**$_{\pm0.1}$ | • Tent+Lana (Ours) | **58.1**$_{\pm0.1}$ |

**Efficiency Evaluation**

Table 9: Efficiency comparisons with SOTA TTA methods by A5000 with VitBase-LN under online imbalanced label shifts on ImageNet-C.

| Method | Running time (hours) |
|---|---|
| • Tent | 2.87 |
| • EATA | 3.08 |
| • AdaContrast | 8.26 |
| • SAR | 5.67 |
| • Tent+Lana (Ours) | 4.39 |

## Integration with other TTA methods

Table 10: Comparisons with SOTA on ImageNet-C (severity level 5) by test accuracy (%) under **Batch Size = 1**. "BN"/"GN"/"LN" denote the Batch/Group/Layer normalization.

| | Gauss | Shot | Impulse | Defocus | Glass | Motion | Zoom | Snow | Frost | Fog | Bright | Contrast | Elastic | Pixel | JPEG |
|---|---|---|---|---|---|---|---|---|---|---|---|---|---|---|---|
| VitBase (LN) | 9.5 | 6.7 | 8.2 | 29.0 | 23.4 | 33.9 | 27.1 | 15.9 | 26.5 | 47.2 | 54.7 | 44.1 | 30.5 | 44.5 | 47.8 |
| • Tent | 42.2 | 1.0 | 43.3 | 52.4 | 48.2 | 55.5 | 50.5 | 16.5 | 16.9 | 66.4 | 74.9 | 64.7 | 51.6 | 67.0 | 64.3 |
| • Tent+Lana (Ours) | **49.0** | **47.6** | **49.6** | **55.3** | **53.1** | **59.2** | **55.4** | **61.0** | **51.1** | **70.3** | **76.7** | **66.7** | **61.1** | **70.2** | **67.6** |
| • EATA | 29.7 | 25.1 | 34.6 | 44.7 | 39.2 | 48.3 | 42.4 | 37.5 | 45.9 | 60.0 | 65.9 | 61.2 | 46.4 | 58.2 | 59.6 |
| • EATA+Lana(Ours) | 31.3 | 28.6 | 35.9 | 46.2 | 45.3 | 51.5 | 46.5 | 46.3 | 46.5 | 65.2 | 69.3 | 64.5 | 50.3 | 62.6 | 64.1 |
| • SAR | 40.8 | 36.4 | 41.5 | 53.7 | 50.7 | 57.5 | 52.8 | 59.1 | 50.7 | 68.1 | 74.6 | 65.7 | 57.9 | 68.9 | 65.9 |
| • SAR+Lana(Ours) | 43.2 | 38.2 | 43.7 | 54.5 | 51.8 | 58.9 | 53.9 | 60.7 | 49.2 | 69.9 | 75.3 | 64.5 | 59.6 | 69.2 | 66.3 |

## Evaluation of different methods under a standard TTA setting

To assess the effectiveness of various TTA approaches under the standard TTA setting, we follow the setup outlined in (Yuan et al., 2024). The results are presented in Table 11.

Table 11: Comparison of Different TTA methods under a standard TTA setting

| Method | Acc (%) ↑ | mCE (%) ↓ |
|---|---|---|
| TENT | 81.41 | 48.13 |
| ETA | 79.58 | 52.64 |
| EATA | 79.59 | 52.62 |
| SAR | 79.77 | 51.94 |
| TEA | 83.34 | 43.69 |
| TEA + Lana | **84.95** | **41.87** |

## Integration with other TTA methods

Table 12: Accuracy comparison of different methods with and without Lana.

| Method | Accuracy (%) |
|---|---|
| Tent | 47.7 |
| Tent + Lana | **59.6** |
| EATA | 46.6 |
| EATA + Lana | **55.3** |
| SAR | 56.3 |
| SAR + Lana | **58.9** |

# F  ALGORITHM

---
**Algorithm 1** *Lana* for TTA.

---
1: **REQUIRE:** pre-trained model parameters $\boldsymbol{\theta}_*$, TTA model learning rate $\eta$, unlearning rate $\alpha$.
2: **for** $k = 1$ to $K$ **do**
3:     Randomly sample a mini-batch of test data $\boldsymbol{x}$
4:     **for** $j = 1$ to $J$ **do**
5:         $\boldsymbol{\theta}_k^j = \boldsymbol{\theta}_k^{j-1} + \alpha[F^{-1}\nabla_{\boldsymbol{\theta}} H(g_{\boldsymbol{\theta}_*}(\boldsymbol{x}))]$ (unlearning)
6:     **end for**
7:     $\boldsymbol{\theta}_{k+1} = \boldsymbol{\theta}_k - \eta\nabla H(g_{\boldsymbol{\theta}_k^j}(\boldsymbol{x}))$ (adapt to test data)
8: **end for**

---

