# OpenReview forum: "An Unlearning-Enhanced General Framework for Test-Time Adaptation"
_ICLR.cc/2026/Conference — ICLR 2026 Conference Withdrawn Submission_

### Official Review · Reviewer_Csov · 2025-10-25

**Soundness:** 2
**Presentation:** 2
**Contribution:** 2
**Rating:** 4
**Confidence:** 3

**Summary:**

This paper proposes a general optimization framework for test-time adaptation (TTA) that unifies existing methods under a Bregman divergence formulation. Building on this framework, the authors introduce Lana, a two-stage method that first "unlearns" source-specific knowledge using the Fisher Information Matrix (FIM) and then adapts to the target domain via entropy minimization. The method is designed to be plug-and-play and is evaluated on ImageNet-C and DomainNet-126 under various distribution shift scenarios. Experimental results show consistent improvements over strong baselines such as Tent, SAR, and EATA.

**Strengths:**

The proposed method achieves noticeable and consistent accuracy gains across multiple challenging TTA scenarios, including label imbalance, small batch sizes, and mixed corruption types.
Introducing 'unlearning' into TTA and proposing a two-stage adaptation process (forgetting first, then adapting) has certain biological inspiration and novelty.
Lana can be integrated into existing TTA methods without modifying their original structure, which enhances its practical appeal.
The general framework based on Bregman divergences offers a unified view of diverse TTA objectives, which may facilitate future theoretical analysis and method development.

**Weaknesses:**

1. The proposed Lana method essentially concatenates two well-studied components: (i) Fisher-weighted parameter updating (borrowed directly from EATA) and (ii) standard entropy minimization (Tent). The resulting pipeline therefore feels like a "stitched" baseline rather than a principled advance, and the observed gains, are better viewed as incremental ablations than conceptual breakthroughs.

2. The combination of different TTA method seem to be exaggerate. The paper simultaneously advertises a "general framework", a "biologically-inspired forgetting mechanism", and a "plug-and-play enhancement". These multiple, partially conflicting narratives obscure the true contribution: a two-stage heuristic that re-weights gradients by the inverse Fisher diagonal. A crisper focus on this practical detail would have been preferable to the sweeping claims of universality.

3. Your comparison methods are outdated and insufficient, and more recent SOTA methods need to be compared. Moreover, most of the methods used in your approach are existing and seem to have not undergone any changes, just being directly applied.

4. Dozens of equations (Taylor expansions, matrix inverses, PAC-Bayesian bounds) are devoted to a one-line update rule. The dense formula load therefore appears ornamental and hinders rapid comprehension and make the description very confusing.

5. The manuscript provides no visual evidence (e.g., t-SNE, and only one figure through the whole article) indicating what knowledge is actually erased or what.

**Questions:**

See the weakness above.

---

### Official Review · Reviewer_1wNH · 2025-10-26

**Soundness:** 2
**Presentation:** 2
**Contribution:** 3
**Rating:** 4
**Confidence:** 3

**Summary:**

The authors propose a general framework for test-time adaptation (TTA) that establishes a unified conceptual foundation for understanding existing approaches as specific instances within a broader optimization perspective. This framework also offers practical guidance for designing new TTA algorithms. Building upon it, the authors introduce a novel method termed Unlearning-Enhanced Test-Time Adaptation (Lana), which adaptively unlearns irrelevant source-domain knowledge before adapting to the target test domain. Comprehensive theoretical analysis and extensive empirical evaluations demonstrate the effectiveness of the proposed method in improving TTA performance.

**Strengths:**

1. This paper presents a general optimization framework for test-time adaptation (TTA), which provides a unified guideline for developing new TTA methods.
2. Building upon this framework, the paper proposes an unlearning-enhanced TTA approach that improves adaptability to target test data distributions.

**Weaknesses:**

1. The authors claim to compare their method with Tent, EATA, AdaContrast, SAR, DeYO, and TEA in the baseline section. However, in many result tables (e.g., Table 2, Table 3, and others), these baselines are not consistently included across all experimental settings. Could the authors clarify the reason for this inconsistency?
2. In the section "Evaluation of different methods under a standard TTA setting," the authors only state that "we follow the setup outlined in (Yuan et al., 2024)". However, the paper does not provide sufficient details about this setup. The results shown in Table 11 are limited and do not convincingly demonstrate Lana’s effectiveness under the standard TTA setting. I suggest the authors include more experiments to support this claim.
3. In Table 3, the bold numbers are presumably meant to highlight specific results. Could the authors clarify what the bold formatting represents? In particular, why are no numbers bolded under the ResNet-50 (BN) setting?
4. In Table 9, the authors only report the runtime of Tent + Lana. I recommend also reporting the runtime when combining Lana with other baselines for consistency. In addition, could the authors provide an analysis of the additional memory overhead introduced by Lana?

**Questions:**

See weaknesses.

---

### Official Review · Reviewer_uE2V · 2025-10-26

**Soundness:** 2
**Presentation:** 3
**Contribution:** 2
**Rating:** 2
**Confidence:** 4

**Summary:**

The paper first introduces a unified Test-Time Adaptation (TTA) optimization framework based on Bregman divergence, which integrates multiple existing paradigms including entropy minimization, pseudo-labeling, weight/output regularization, and Bayesian approaches into one general formulation.Building on this, it proposes Lana, a two-step TTA method: first, it performs selective unlearning; second, it conducts standard target-domain adaptation.
Main contribution:This paper presents a unified theoretical framework for Test-Time Adaptation (TTA) and introduces a new paradigm of “unlearning first, then adapting” (Lana). It provides an update rule that can be easily combined with existing TTA methods, and demonstrates superior performance over multiple state-of-the-art (SOTA) approaches on benchmarks such as ImageNet-C.

**Strengths:**

1. Unifying perspective: The Bregman-based objective offers a tidy lens to rewrite several existing methods.
2. Method simplicity & implementability: Lana is easy to plug in (one-shot, diagonal FIM; no source labels), and can be combined with common baselines such as Tent with minimal code changes.Estimating F once and using a diagonal approximation keeps the overhead low.
3. Experimental stress tests: Includes difficult ImageNet-C regimes (severity 5, label imbalance, bs=1, mixed corruptions), which are relevant stressors for TTA.
4. Clarity of presentation: Key equations are explicit; the special-case reductions and algorithmic flow are reasonably easy to follow.

**Weaknesses:**

1. The current Bregman-divergence-based unification appears mostly formal, lacking justification for why such unification is necessary and what concrete benefits it brings. The paper should present new variants systematically derived from the framework that show empirical advantages, and offer unified theoretical insights (e.g., on stability, flatness, or convergence) to demonstrate added value rather than mere reformulation.
2. The “unlearning-then-adaptation” idea is appealing, yet the implementation essentially reduces to a Fisher-preconditioned entropy gradient step. This is closely related to EWC, EATA, natural gradient, or generalized Gauss–Newton methods, making the innovation limited. The link between the conceptual motivation (“selective unlearning”) and the concrete method is weak. The authors should clarify why this specific step embodies unlearning and how it fundamentally differs from existing Fisher-based approaches; otherwise, the novelty risks being seen as a repackaging of known techniques.
3. The comparisons focus mainly on classical methods, lacking evaluation against more recent and stronger baselines. Without up-to-date SOTA results under consistent experimental protocols, the claimed improvements remain unconvincing.
4. Theoretical approximation validity and error bounds unclear. The method relies on first- and second-order Taylor expansions, yet the paper does not specify the magnitude of the neglected higher-order terms or the radius of validity of these approximations.

**Questions:**

1. The proposed general TTA optimization framework is based on Bregman divergence.  Please clarify the unique theoretical or practical advantages of this formulation. For example, can it systematically inspire new algorithmic designs or provide novel insights into stability or generalization?
2. The core of the proposed method appears nearly equivalent to a single Fisher-preconditioned update. Please explain how this process mechanistically achieves “selective unlearning,” and in what essential ways Lana differs from other Fisher-based approaches such as EWC, EATA, or natural gradient methods.
3. In the paper, the Fisher Information Matrix F is computed only once and approximated diagonally. If F changes over time, would the algorithm’s performance degrade? Have you considered dynamic re-estimation or damping mechanisms?

---

### Official Review · Reviewer_VVW5 · 2025-11-01

**Soundness:** 3
**Presentation:** 2
**Contribution:** 2
**Rating:** 6
**Confidence:** 5

**Summary:**

This paper proposes a general optimization framework for Test-Time Adaptation (TTA) using Bregman divergences to unify entropy minimization, pseudo-labeling, weight/output regularization, and Bayesian methods. It reveals a key limitation: existing TTA assumes all source knowledge is beneficial. The authors introduce Lana, an unlearning-enhanced method that first adaptively removes irrelevant source knowledge via Fisher-based parameter importance, then adapts to the target domain.

**Strengths:**

1. Lana is a principled extension: unlearning irrelevant source knowledge via Fisher inverse addresses negative transfer — biologically and empirically motivated.

2. Strong empirical gains: +13.8% over Tent on ImageNet-C (ViT), +15.4% on ResNet50; consistent across corruptions and natural shifts.

3. Thorough ablations validate unlearning threshold, integration with multiple TTA baselines, and robustness to batch size/class imbalance.

**Weaknesses:**

1. The proposed unified framework is more of a summary and categorization of prior work, grouping loss functions via hyperparameters and Bregman divergences — serves only a preliminary role, not a core contribution. Authors should invest more in deeply exploring machine unlearning’s role in TTA with theory and experiments.

2. How is F (Fisher matrix) computed? Diagonal approximation is used, but no details on empirical Fisher vs. true Fisher, stability, or computational cost.

3. Equations (9)–(11) are vaguely explained — purpose of Taylor expansions and their role in unlearning/adaptation is not clearly introduced.

4. Why does $ F^{-1} $ indicate parameter importance for source domain knowledge? Intuition is stated but not rigorously justified; connection to forgetting is weak.

**Questions:**

Please refer to Weaknesses.

---

### Note · Authors · 2025-11-14

I have read and agree with the venue's withdrawal policy on behalf of myself and my co-authors.